# DECENTRALIZED COOPERATIVE MULTI-AGENT REINFORCEMENT LEARNING WITH EXPLORATION

## ABSTRACT

Many real-world applications of multi-agent reinforcement learning (RL), such as multi-robot navigation and decentralized control of cyber-physical systems, involve the cooperation of agents as a team with aligned objectives. We study multi-agent RL in the most basic cooperative setting — Markov teams — a class of Markov games where the cooperating agents share a common reward. We propose an algorithm in which each agent independently runs stage-based V-learning (a Q-learning style algorithm) to efficiently explore the unknown environment, while using a stochastic gradient descent (SGD) subroutine for policy updates. We show that the agents can learn an $\varepsilon$-approximate Nash equilibrium policy in at most $\propto \widetilde{O}(1/\varepsilon^4)$ episodes. Our results advocate the use of a novel *stage-based* V-learning approach to create a stage-wise stationary environment. We also show that under certain smoothness assumptions of the team, our algorithm can achieve a nearly *team-optimal* Nash equilibrium. Simulation results corroborate our theoretical findings. One key feature of our algorithm is being *decentralized*, in the sense that each agent has access to only the state and its local actions, and is even *oblivious* to the presence of the other agents. Neither communication among teammates nor coordination by a central controller is required during learning. Hence, our algorithm can readily generalize to an arbitrary number of agents, without suffering from the exponential dependence on the number of agents.

## 1 INTRODUCTION

A variety of real-world sequential decision-making problems have been successfully addressed with reinforcement learning (RL), such as playing the game of Go (Silver et al., 2016), Poker (Brown & Sandholm, 2018), real-time strategy games (Vinyals et al., 2019), autonomous driving (Shalev-Shwartz et al., 2016), and robotics (Kober et al., 2013). Many of these applications can be cast as a multi-agent reinforcement learning (MARL) problem, where multiple agents are involved in an interactive environment. These successful applications have inspired a surging interest in understanding the theoretical aspects of MARL, which, however, is mostly devoted to the competitive setting, e.g., two-player zero-sum Markov games. This has left many problems in other important MARL settings, especially cooperative MARL, relatively open.

Cooperative MARL generally involves a team of agents collaborating on a common task. We roughly categorize cooperative MARL settings into three classes based on the *information* available to the agents. The first class is (1) centralized/joint learning (Boutilier, 1996; Claus & Boutilier, 1998), which assumes that there exists a single coordinator who can access the local information of all the agents, and makes decisions/learns policies jointly for all of them. This centralized training (though possibly decentralized execution) approach has become a common practice in empirical MARL (Oliehoek et al., 2008; Foerster et al., 2016; Lowe et al., 2017; Rashid et al., 2018; Son et al., 2019). Centralized learning essentially reduces the multi-agent problem to a single-agent one, but such a method does not scale well with the number of agents. In fact, one can show that the computation & sample complexity grows exponentially with the number of agents in the worst case. The second class is (2) learning with communications (Kar et al., 2013; Zhang et al., 2018; Dubey & Pentland, 2021), which typically assumes that the agents can communicate by sharing some local information with other agents (sometimes through a central agent). This approach gets around the centralized computation bottleneck, but instead suffers from additional communication overhead. It can also be

unrealistic in some real-world scenarios where communication may be expensive and/or unreliable, such as in unmanned aerial vehicle (UAV) field coverage (Pham et al., 2018).

Given the aforementioned limitations, we are more interested, in this paper, in a more practical setting: (3) *decentralized* learning[1]. We enforce a seemingly daunting restriction: Each agent only has access to its own local information (e.g., local actions and rewards), and can neither communicate with its teammates nor be coordinated by any central controller during learning. In fact, we require that each agent is completely unaware of the underlying structure of the game (whether it is cooperative or not), or even the presence of other agents. Among the three classes mentioned above, decentralized learning requires the weakest assumption on information availability, which makes it suitable for many practical multi-agent learning scenarios (Fudenberg et al., 1998; Wang & de Silva, 2008; Kuyer et al., 2008). In addition, decentralized learning is generally more scalable, and does not suffer from the exponential sample & computation complexity in the number of agents. Seemingly tempting, one might wonder how strong, if any, theoretical results can be established in this challenging decentralized setting. In the paper, we investigate the theoretical aspects of decentralized cooperative multi-agent RL in the *non-asymptotic* regime. A natural and fundamental theoretical question is:

*Does any independent Q-learning style algorithm lead to Nash equilibria in cooperative MARL?*

The answer to this question has remained elusive ever since it was raised in Lauer & Riedmiller (2000). To answer this question, we focus on arguably the most basic and fundamental cooperative MARL setting — Markov teams (Lauer & Riedmiller, 2000; Boutilier, 1996; Wang & Sandholm, 2002) – where the cooperating agents share a common team-reward function, and the state transition in the environment is affected by the joint actions of all agents. In fact, even in the basic Markov team setting, decentralized learning can be highly challenging. First, since both rewards and transitions are affected by the other agents, the environment becomes *non-stationary* from each agent's own perspective when the other agents also update their policies. Hence, an agent needs to explore the unknown environment efficiently while keeping in mind that the information it gathers now will soon become outdated. This rules out many successful single-agent RL solutions, which critically rely on the assumption that the agent is learning in a stationary environment. Second, compared with RL in two-player zero-sum games, an additional challenge in teams is the existence of possibly multiple Nash equilibria (NE) with different values. The agents in teams hence have to additionally coordinate on which NE to converge to. Our contributions in the paper are summarized as follows.

**Contributions.** 1) As a warm-up, we start with decentralized matrix teams (Section 3), which constitute a special class of Markov teams with no state transitions. We show that a simple stochastic gradient descent (SGD) algorithm can provably find an $\varepsilon$-approximate NE policy using $\propto O(1/\varepsilon^8)$ samples, which can be improved to $\propto \widetilde{O}(1/\varepsilon^4)$ by using variance reduction techniques. 2) For the general problem of decentralized Markov teams (Section 4), we propose an algorithm named Stage-Based V-learning with Stochastic Gradient Descent (`V-learning SGD`), where V-learning (a name first coined in Bai et al. (2020)) is a simple variant of Q-learning. Our algorithm can be executed in a decentralized way as we defined above. 3) We show in Section 5 that if all agents in the team run `V-learning SGD`, they can find an $\varepsilon$-approximate NE in at most $\widetilde{O}(1/\varepsilon^4) \cdot \text{poly}(S, H, A_{\max})$ episodes, where $S$ is the number of states, $A_{\max}$ is the size of the largest action space, and $H$ is the length of an episode. Our results rely on a novel investigation of *stage-based* V-learning to create a stage-wise stationary environment for the agents. 4) We further show that our algorithm can approach the best equilibrium, i.e., the team-optimality, in a specific class of problems named "smooth teams". 5) In Section 6, we provide numerical results that corroborate our theoretical findings. To the best of our knowledge, our work appears to be the first to provide non-asymptotic guarantees for cooperative MARL with exploration, while having the additional advantage of being decentralized.

**Related Work.** A common mathematical framework of multi-agent RL is stochastic games (Shapley, 1953), which is also referred to as Markov games. Early attempts to learn NE in Markov games include Littman (1994; 2001); Hu & Wellman (2003); Hansen et al. (2013), but they either assume the transition kernel and rewards are known, or only yield asymptotic guarantees. Recently, various sample efficient methods have been proposed (Wei et al., 2017; Bai & Jin, 2020; Sidford et al., 2020;

---

[1]This setting has been studied under various names in the literature, including individual learning (Leslie & Collins, 2005), decentralized learning (Arslan & Yüksel, 2016), agnostic learning (Tian et al., 2021; Wei et al., 2021), and independent learning (Claus & Boutilier, 1998; Daskalakis et al., 2020). It also belongs to a more general category of teams/games with decentralized information structure (Ho, 1980; Nayyar et al., 2013a;b).

Xie et al., 2020; Bai et al., 2020; Liu et al., 2021; Zhao et al., 2021; Guo et al., 2021), mostly for learning in two-player zero-sum Markov games. Several works have investigated zero-sum games in a *decentralized* setting as we consider here (Daskalakis et al., 2020; Tian et al., 2021; Wei et al., 2021; Sayin et al., 2021), but these results do not carry over in any way to the decentralized cooperative/team setting. We refer the reader to Appendix A for a more detailed discussion on these related works.

Another line of research has considered RL in teams. Without enforcing a decentralized environment, Boutilier (1996) has proposed to coordinate the agents by letting them take actions in a lexicographic order. Wang & Sandholm (2002) have studied optimal adaptive learning that converges to the optimal NE in Markov teams. These two methods critically rely on communications among the agents (beforehand) and/or observing the teammates' actions. In contrast, the distributed Q-learning algorithm in Lauer & Riedmiller (2000) is decentralized and coordination-free, which, however, only works for *deterministic* tasks, and has no non-asymptotic convergence guarantees. In fact, developing provable decentralized RL for cooperation in general *stochastic* environments is considered an important open problem in Lauer & Riedmiller (2000), which we address in this paper.

Arslan & Yüksel (2016) and Yongacoglu et al. (2019) have shown that decentralized Q-learning style algorithms can converge to NE in weakly acyclic games, which cover Markov teams as an important special case. Their decentralized settings are most similar to ours in that each agent is oblivious to the presence of the others. However, both of them require a coordinated exploration phase, and only yield asymptotic guarantees. In fact, it is not clear if their methods will converge in polynomial time. We significantly improve their results in this respect, by explicitly showing a polynomial sample complexity upper bound. Two contemporaneous works (Zhang et al., 2021b; Leonardos et al., 2021) have studied gradient play in Markov potential games,[2] but they do not consider the aspect of *exploring* the unknown environment, one of the key issues in RL, and/or have to assume perfect knowledge of the environment. A concurrent work (Chang et al., 2021) has studied cooperative multi-player multi-armed bandits with information asymmetry, which parallels our matrix team setting. Nevertheless, (Chang et al., 2021) requires stronger conditions than our decentralized setting as their algorithm relies on playing a predetermined sequence of actions.

## 2 Preliminaries

An $N$-player episodic Markov team is defined by a tuple $(\mathcal{N}, H, \mathcal{S}, \{\mathcal{A}^i\}_{i=1}^N, r, P)$, where (1) $\mathcal{N} = \{1, 2, \dots, N\}$ is the set of agents; (2) $H \in \mathbb{N}_+$ is the number of time steps in each episode; (3) $\mathcal{S}$ is the finite state space; (4) $\mathcal{A}^i$ is the finite action space for agent $i \in \mathcal{N}$; (5) $r : [H] \times \mathcal{S} \times \mathcal{A} \to [0, 1]$ is the team reward function for all agents, where $\mathcal{A} = \times_{i=1}^N \mathcal{A}^i$ is the joint action (or action profile) space; and (6) $P : [H] \times \mathcal{S} \times \mathcal{A} \to \Delta(\mathcal{S})$ is the transition kernel. We remark that both the reward function and the state transition function depend on the joint actions of all the agents.

The agents interact in an unknown environment for $K$ episodes. At each time step $h \in [H]$, the agents observe the state $s_h \in \mathcal{S}$, and take actions $a_h^i \in \mathcal{A}^i, i \in \mathcal{N}$ simultaneously. All agents receive the same reward $r_h(s_h, a_h)$, where $a_h = (a_h^1, \dots, a_h^N)$, and the environment transitions to the next state $s_{h+1} \sim P_h(\cdot|s_h, a_h)$. We assume for simplicity that the initial state $s_1$ of each episode is fixed. Note that the state transition here is general and not restricted to be deterministic, in contrast to Lauer & Riedmiller (2000). This makes decentralized learning considerably more challenging, as the agents cannot implicitly coordinate by enumerating/rehearsing all possible states. We focus on the *decentralized* setting, where each agent only observes the states, rewards, and its own actions, but not the actions of the other agents. We require that each agent is completely oblivious of the existence of the others, and is also not allowed to communicate. This decentralized information structure requires each agent to learn to make decisions based on only its local information.

For each agent $i$, a (Markov) policy $\pi^i : [H] \times \mathcal{S} \to \Delta(\mathcal{A}^i)$ is a mapping from the time index and state space to a distribution over its own action space. Each agent seeks to find a policy that maximizes its own reward. A joint policy (or policy profile) $\pi = (\pi^1, \dots, \pi^N)$ induces a probability measure over the sequence of states and joint actions. For convenience, we use the superscript $-i$ to denote the set of agents excluding agent $i$, i.e., $\mathcal{N}\setminus\{i\}$. For example, we can rewrite $\pi = (\pi^i, \pi^{-i})$

---

[2] All our results in this paper immediately generalize to Markov potential games, which include Markov teams as a special case. For simplicity of notations, we illustrate our algorithm and results only in the Markov team setting. A short discussion on Markov potential games is included in Appendix B.

using this convention. For a policy profile $\pi$, and for any $h \in [H]$, $s \in \mathcal{S}$, and $a \in \mathcal{A}$, we define the value function and the state-action value function (or $Q$-function) as follows:

$$V_h^\pi(s) \overset{\text{def}}{=} \mathbb{E}_\pi \Big[ \sum_{h'=h}^H r_{h'}(s_{h'}, a_{h'}) \mid s_h = s \Big], Q_h^\pi(s,a) \overset{\text{def}}{=} \mathbb{E}_\pi \Big[ \sum_{h'=h}^H r_{h'}(s_{h'}, a_{h'}) \mid s_h = s, a_h = a \Big]. \quad (1)$$

For agent $i$, a policy $\pi^{i\star}$ is a *best response* to $\pi^{-i}$ if $V_h^{\pi^{i\star}, \pi^{-i}}(s) = \sup_{\pi^i} V_h^{\pi^i, \pi^{-i}}(s)$, for any step $h \in [H]$ and state $s \in \mathcal{S}$. A policy profile $\pi = (\pi^i, \pi^{-i})$ is a Nash equilibrium if $\pi^i$ is a best response to $\pi^{-i}$ for all $i \in \mathcal{N}$. One can show that NE always exists in finite Markov teams (Boutilier, 1996). There can be multiple NE in a Markov team with possibly different values. One desirable equilibrium is the team-optimal one that achieves the highest value. Specifically, a policy profile $\pi^\star$ is *team-optimal* if $V_h^{\pi^\star}(s) = \sup_\pi V_h^\pi(s)$, for any step $h \in [H]$ and state $s \in \mathcal{S}$. It follows immediately that a team-optimal policy profile is always a NE, but the converse is not true in general.

For general (non-Markov) policies, we can still define their values at step $h = 1$ in a sense similar to Equation (1). A best response $\pi^{i\star}$ with respect to general policies $\pi^{-i}$ maximizes player $i$'s value at step 1, i.e., $V_1^{\pi^{i\star}, \pi^{-i}}(s_1) = \sup_{\pi^i} V_1^{\pi^i, \pi^{-i}}(s_1)$. The best response to the general policies of the opponents is not necessarily Markov. We also have an approximate notion of Nash equilibrium:

**Definition 1.** *($\varepsilon$-approximate Nash equilibrium). For $\varepsilon > 0$, a general policy profile $\pi = (\pi^i, \pi^{-i})$ is an $\varepsilon$-approximate Nash equilibrium if $V_1^{\pi^i, \pi^{-i}}(s_1) \geq \sup_{\pi^{i'}} V_1^{\pi^{i'}, \pi^{-i}}(s_1) - \varepsilon$, for all $i \in \mathcal{N}$, where the supremum is taken over all general policies that are not necessarily Markov.*

For notational convenience, in most parts of the paper we illustrate our algorithms and results for two-player teams, i.e., $N = 2$. It is straightforward to extend our results to the general $N$-player team as we defined above. With two players, we use $\mathcal{A}$ and $\mathcal{B}$ to denote the action spaces of players 1 and 2, respectively. Let $S = |\mathcal{S}|, A = |\mathcal{A}|, B = |\mathcal{B}|$. Denote by $A_{\max} = \max\{A, B\}$, which is known to both players. We also rewrite the policies $(\pi^1, \pi^2)$ as $(\mu, \nu)$ for notational convenience.

## 3 WARM-UP: THE MATRIX TEAM

As a warm-up, we start with a simple matrix team problem, where two agents repeatedly interact in a single-stage team (Claus & Boutilier, 1998) with stochastic rewards. This is essentially a Markov team with a single state and $H = 1$. The intuitions and results in this section will become useful later in our analysis of the general Markov team problem.

A two-player finite matrix team can be described by a tuple $(\mathcal{A}, \mathcal{B}, \rho)$, where $\mathcal{A}$ and $\mathcal{B}$ are the finite action spaces for agent 1 and agent 2, respectively, and the stochastic reward function $\rho = (P_{a,b} : a \in \mathcal{A}, b \in \mathcal{B})$ is a collection of reward distributions with support $[0, 1]$. The agents interact for $T$ rounds. At each round $t \in [T]$, agent 1 (resp. agent 2) chooses an action $a_t \in \mathcal{A}$ (resp. $b_t \in \mathcal{B}$). Both agents receive the same reward $R_t$ drawn i.i.d. from the reward distribution $P_{a_t, b_t}$. We let $r : \mathcal{A} \times \mathcal{B} \to [0, 1]$ be the expected value of the reward function, i.e., $r(a, b) = \int_0^1 x dP_{a,b}(x)$. We use $\mu \in \Delta(\mathcal{A})$ and $\nu \in \Delta(\mathcal{B})$ to denote the *mixed strategies* (or simply strategies) for agent 1 and agent 2, respectively. Slightly abusing notations, the value function for a strategy pair $\pi = (\mu, \nu)$ is denoted by $V(\pi) = V(\mu, \nu) \overset{\text{def}}{=} \mathbb{E}_{\mu \times \nu}[r(a, b)]$, where $a \sim \mu$ and $b \sim \nu$.

In an unknown environment, a natural tactic for a self-interested agent is to adopt a no-regret learning algorithm to guide its action selection (Cesa-Bianchi & Lugosi, 2006). In our problem, we propose to let each agent run a simple no-regret learning algorithm, namely (projected) stochastic gradient descent (SGD). Our Stochastic Gradient Descent with Implicit Exploration (`SGD-IX`) algorithm for agent 1 is presented in Algorithm 1. The algorithm run by agent 2 is symmetric and hence not included. Our use of SGD is motivated by the simple observation that every first-order stationary point of the value function is also a NE for the matrix team (Lemma 4), and that SGD does lead to stationary points in non-convex optimization (Ghadimi & Lan, 2013). Our solution also relies on the fact that centralized SGD can be decoupled into a decentralized one (Lemma 5), as the joint strategy is a Cartesian product of individual strategies. For generality, we write our gradient descent step in Algorithm 1 as an online mirror descent (OMD) update. For more detailed descriptions and intuitions of Algorithm 1, we refer the reader to Appendix D. The following result shows that the output strategy pair of Algorithm 1 is close to a NE in expectation.

---

**Algorithm 1:** `SGD-IX`: Stochastic Gradient Descent with Implicit Exploration (for player 1)

---

1 **Initialize:** $\mu_{1,a} \leftarrow 1/A, \forall a \in \mathcal{A}, \ \alpha_t \leftarrow T^{-3/4}, \forall t \in [T]$, and $\gamma \leftarrow T^{-1/4}$.
2 **for** $t \leftarrow 1$ *to* $T$ **do**
3      Draw action $a_t \sim \mu_t = (\mu_{t,1}, \ldots, \mu_{t,A})$;
4      Take action $a_t$, and observe the team reward $R_t$;
5      $\widetilde{\nabla}_{\mu,a} L_t \leftarrow \frac{1 - R_t}{\mu_{t,a} + \gamma} \mathbb{I}\{a_t = a\}, \forall a \in \mathcal{A}$, and $\widetilde{\nabla}_\mu L_t \leftarrow (\widetilde{\nabla}_{\mu,1} L_t, \ldots, \widetilde{\nabla}_{\mu,A} L_t)$;
6      $\mu_{t+1} \leftarrow \operatorname{argmin}_{\mu \in \Delta(\mathcal{A})} \left\{ \langle \widetilde{\nabla}_\mu L_t, \mu \rangle + \frac{1}{\alpha_t} D_\omega(\mu, \mu_t) \right\}$, where $\omega(\mu) = \frac{1}{2} \|\mu\|_2^2$, and
     $D_\omega(\mu, \mu') = \omega(\mu) - \omega(\mu') - \langle \nabla \omega(\mu'), \mu - \mu' \rangle$;
7 Output $\mu_\tau$ where $\tau$ is uniformly sampled from $\{1, \ldots, T\}$;

---

**Theorem 1.** *Suppose that both agents run Algorithm 1 for $T$ rounds. Let the two agents uniformly draw a time index $\tau$ from $\{1, \ldots, T\}$, and denote by $x_\tau = (\mu_\tau, \nu_\tau)$ the output of the algorithm. For any $\varepsilon > 0$, if $T = \tilde{\Omega}(1/\varepsilon^8) \cdot poly(A_{\max})$, then $(\mu_\tau, \nu_\tau)$ is an $\varepsilon$-approximate Nash equilibrium in expectation, where $poly(A_{\max})$ denotes a polynomial of $A_{\max}$.*

Note that to extract an approximate NE strategy, the agents need to sample the same time index $\tau$. This requires some common source of randomness like a shared random seed. Such common randomness is also termed a correlation device, and is standard in decentralized learning (Bernstein et al., 2009; Arabneydi & Mahajan, 2015; Zhang et al., 2019). Note that the correlation device is never used during the learning process to coordinate the exploration, but is simply used to synchronize the selection of the strategies after they have already been generated by the learning algorithm. A common random seed is generally considered as a mild assumption and does not break the decentralized paradigm. In addition, we would like to remark that Theorem 1 is not an immediate consequence of, and, in fact, is fundamentally different from, the folklore result that the empirical frequency of plays of independent no-regret learning converges to the set of coarse correlated equilibria (CCE) in general-sum normal-form games (Hart & Mas-Colell, 2000). In teams, the set of NE can be a strict subset of CCE in general. Theorem 1 also crucially takes advantage of the first-order stationarity of NE in teams, a property that is clearly absent in the folklore result of general-sum games.

In the following theorem, we further show that the sample complexity of learning in matrix teams can be improved to $\tilde{O}(1/\varepsilon^4)$ by utilizing a variance reduction technique. The corresponding algorithm and its analysis are deferred to the appendices due to space limitations.

**Theorem 2.** *Let the agents run variance-reduced SGD (Algorithm 4 in Appendix E) for $T$ rounds. For any $\varepsilon > 0$, if $T = \tilde{\Omega}(1/\varepsilon^4) \cdot poly(A_{\max})$, then $(\mu_\tau, \nu_\tau)$ is an $\varepsilon$-approximate NE in expectation.*

## 4   A DECENTRALIZED COOPERATIVE MARL ALGORITHM

In this section, we introduce our algorithm Stage-Based V-learning with Stochastic Gradient Descent (`V-learning SGD`) for decentralized multi-agent RL in Markov teams. A well-known key challenge in MARL is that the environment becomes non-stationary from each agent's own perspective when the other agents also update their policies (Busoniu et al., 2008; Zhang et al., 2021a). To address such a challenge, we propose to use a *stage-based* Q-learning style algorithm to create a stage-wise stationary environment for the agents, while invoking the `SGD-IX` subroutine (Algorithm 1) in each such stationary stage for policy updates. As an interesting side remark, stage-based Q-learning has also achieved near-optimal regret bounds in single-agent RL (Zhang et al., 2020b).

The algorithm run by agent 1 is presented in Algorithm 2. The algorithm for agent 2 is symmetric, by simply replacing the action space $\mathcal{A}$ with $\mathcal{B}$. Each agent runs a stage-based V-learning algorithm independently, and maintains upper confidence bounds on the value functions to actively explore the unknown environment. For each step-state pair $(h, s)$, we divide the visitations to this pair into multiple *stages*, where the lengths of the stages increase exponentially at a rate of $(1 + 1/H)$ (Zhang et al., 2020b). Specifically, we let $e_1 = H$, and $e_{i+1} = \lfloor (1 + 1/H)e_i \rfloor, i \geq 1$ denote the lengths of the stages, and let the partial sums $\mathcal{L} \stackrel{\text{def}}{=} \{ \sum_{i=1}^j e_i \mid j = 1, 2, 3, \ldots \}$ denote the set of ending times of the stages. For each $(h, s)$ pair, we update our optimistic estimates $\overline{V}_h(s_h)$ of the value function at the end of each stage (that is, when the total number of visitations to $(s, h)$ lies in the set

---

**Algorithm 2:** `V-learning SGD`: Stage-Based V-learning with Stochastic Gradient Descent

---

1    **Initialize:** $\overline{V}_h(s) \leftarrow H - h + 1, N_h(s) \leftarrow 0, \check{N}_h(s) \leftarrow 0, \check{r}_h(s) \leftarrow 0, \check{v}_h(s) \leftarrow 0, \check{T}_h(s) \leftarrow H,$
     and $\mu_h^1(a \mid s) \leftarrow 1/A, \forall h \in [H], s \in \mathcal{S}, a \in \mathcal{A}.$

2    **for** *episode* $k \leftarrow 1$ *to* $K$ **do**

3      Receive $s_1$;

4      **for** *step* $h \leftarrow 1$ *to* $H$ **do**

5         $N_h(s_h) \leftarrow N_h(s_h) + 1, \check{n} \overset{\text{def}}{=} \check{N}_h(s_h) \leftarrow \check{N}_h(s_h) + 1;$

6         Take action $a_h \sim \mu_h^{\check{n}}(\cdot \mid s_h)$, and observe reward $r_h$ and next state $s_{h+1}$;

7         $\check{r}_h(s_h) \leftarrow \check{r}_h(s_h) + r_h, \check{v}_h(s_h) \leftarrow \check{v}_h(s_h) + \overline{V}_{h+1}(s_{h+1});$

8         **if** $N_h(s_h) \in \mathcal{L}$ **then**

9            `//Entering a new stage`

10           $\overline{V}_h(s_h) \leftarrow \frac{\check{r}_h(s_h)}{\check{n}} + \frac{\check{v}_h(s_h)}{\check{n}} + b_{\check{n}}$, where $b_{\check{n}} \leftarrow 25H\sqrt{A_{\max}^3/\check{n}^{1/4}};$

11           $\check{N}_h(s_h) \leftarrow 0, \check{r}_h(s_h) \leftarrow 0, \check{v}_h(s_h) \leftarrow 0, \check{T}_h(s_h) \leftarrow \lfloor (1 + \frac{1}{H})\check{T}_h(s_h) \rfloor;$

12           $\mu_h^1(a \mid s_h) \leftarrow 1/A, \forall a \in \mathcal{A};$

13         $\alpha_{\check{n}} \leftarrow (\check{T}_h(s_h))^{-3/4}, \gamma \leftarrow (\check{T}_h(s_h))^{-1/4};$

14         $\widetilde{\nabla}_{\mu,a} L_{\check{n}} \leftarrow \frac{H-h+1-(r_h+\overline{V}_{h+1}(s_{h+1}))}{\mu_h^{\check{n}}(a \mid s_h) + \gamma} \mathbb{I}\{a_h = a\}, \forall a \in \mathcal{A};$

15         $\widetilde{\nabla}_\mu L_{\check{n}} \leftarrow (\widetilde{\nabla}_{\mu,1} L_{\check{n}}, \ldots, \widetilde{\nabla}_{\mu,A} L_{\check{n}});$

16         $\mu_h^{\check{n}+1}(\cdot \mid s_h) \leftarrow \text{argmin}_{\mu \in \Delta(\mathcal{A})} \left\{ \langle \widetilde{\nabla}_\mu L_{\check{n}}, \mu \rangle + \frac{1}{\alpha_{\check{n}}} D_\omega(\mu, \mu_h^{\check{n}}(\cdot \mid s_h)) \right\}$, where
         $\omega(\mu) = \frac{1}{2}\|\mu\|_2^2$, and $D_\omega(\mu, \mu') = \omega(\mu) - \omega(\mu') - \langle \nabla\omega(\mu'), \mu - \mu' \rangle;$

---

$\mathcal{L}$), using samples only from this single stage. This way, our stage-based V-learning ensures that only the most recent $O(1/H)$ fraction of the collected samples are used to calculate $\overline{V}_h(s_h)$, while the first $1 - O(1/H)$ fraction is forgotten. Such a stage-based update framework in some sense mimics the celebrated optimistic Q-learning algorithm with a learning rate of $\alpha_t = \frac{H+1}{H+t}$ (Jin et al., 2018), which also roughly uses the last $O(1/H)$ fraction of samples for value updates. However, we would like to specifically emphasize that the V-learning variant of Jin et al. (2018) does not directly apply to our problem due to some technical challenges that will be covered in Section 5. Unlike Jin et al. (2018), our stage-based V-learning ensures that the value function estimates are updated at a low frequency, a property also termed *low-switching cost* in single-agent RL (Bai et al., 2019), which is crucial to address the non-stationarity issue in our MARL problem.

Another key component of Algorithm 2 is the SGD subroutine. For each step $h \in [H]$ and each state $s_h \in \mathcal{S}$, we treat the agents' interactions at $s_h$ as a separate matrix team problem, similar to what we have defined in Section 3. The reward function for such a matrix team is the $Q$-function at the corresponding state, which is directly related to the optimistic value function estimates $\overline{V}_h(s_h)$ that we have constructed in the V-learning part. Since our stage-based V-learning only updates $\overline{V}_h(s_h)$ at the end of a stage, it creates a stage-wise stationary environment in the sense that the value of $\overline{V}_h(s_h)$ is fixed within each stage, which can be captured by our stationary problem formulation in Section 3. We can hence directly invoke Algorithm 1 and its analyses for the matrix team problem at each step-state pair $(h, s_h)$. Finally, we remark that both the V-learning and the SGD components of our algorithm are decentralized, which can be implemented using only the states observed and the local actions executed, without any communication or central coordination among the agents.

## 5   Theoretical Analyses

In this section, we present our main results on the sample complexity of `V-learning SGD`. We first introduce a few notations to facilitate the analysis. For a step $h \in [H]$ of an episode $k \in [K]$, we denote by $s_h^k$ the state that the agents observe at this time step. For any state $s \in \mathcal{S}$, we let $\mu_h^k(\cdot \mid s) \in \Delta(\mathcal{A})$ and $\nu_h^k(\cdot \mid s) \in \Delta(\mathcal{B})$ be the strategies prescribed by Algorithm 2 to agents 1 and 2, respectively, at this step. Notice that such notations are well-defined for every $s \in \mathcal{S}$ even if $s$ might not be the state $s_h^k$ that is actually visited. We further let $\mu_h^k = \{\mu_h^k(\cdot \mid s) : s \in \mathcal{S}\}$ and $\nu_h^k = \{\nu_h^k(\cdot \mid s) : s \in \mathcal{S}\}$. Let $a_h^k \in \mathcal{A}$ and $b_h^k \in \mathcal{B}$ be the actual actions taken by the two agents.

---

**Algorithm 3:** Construction of the Output Policy $\bar{\mu}_h^k$

---

1 **Input:** The strategy trajectory $\{\mu_h^k\}_{h=1,k=1}^{H,K}$ specified by Algorithm 2.
2 **Initialize:** $k' \leftarrow k$.
3 **for** *step* $h' \leftarrow h$ *to* $H$ **do**
4      Receive $s_{h'}$;
5      Take action $a_{h'} \sim \mu_{h'}^{k'}(\cdot \mid s_{h'})$;
6      Uniformly sample $i$ from $\{1, 2, \ldots, \check{N}_{h'}^{k'}(s_{h'})\}$;
7      Set $k' \leftarrow \check{l}_{h',i}^{k'}$, where recall that $\check{l}_{h',i}^{k'}$ is the index of the episode such that state $s_{h'}$ was visited the $i$-th time (among the total $\check{N}_{h'}^{k'}(s_{h'})$ times) in the last stage;

---

For any $s \in \mathcal{S}$, let $N_h^k(s)$, $\check{N}_h^k(s)$, and $\overline{V}_h^k(s)$ denote, respectively, the values of $N_h(s)$, $\check{N}_h(s)$, and $\overline{V}_h(s)$ at the *beginning* of the $k$-th episode. Note that it is proper to use the same notation to denote these values from the two agents' perspectives, because both agents maintain the same estimates of these three terms as they can be calculated from the common observations (of the state-visitation).

Further, for a state $s_h^k$, let $\check{n}_h^k$ denote the number of times that state $s_h^k$ has been visited (at the $h$-th step) in the stage right before the current stage, and let $\check{l}_{h,i}^k$ denote the index of the episode that this state was visited the $i$-th time among the $\check{n}_h^k$ times. For notational convenience, we use $\check{n}$ to denote $\check{n}_h^k$, and $\check{l}_i$ to denote $\check{l}_{h,i}^k$, whenever $h$ and $k$ are clear from the context. With the new notations, the update rule in Line 10 of Algorithm 2 can be equivalently expressed as

$$\overline{V}_h(s_h) \leftarrow \frac{1}{\check{n}} \sum_{i=1}^{\check{n}} \left( r_h(s_h, a_h^{\check{l}_i}, b_h^{\check{l}_i}) + \overline{V}_{h+1}^{\check{l}_i}(s_{h+1}^{\check{l}_i}) \right) + b_{\check{n}}. \tag{2}$$

Similar to the situation of learning in zero-sum Markov games (Bai et al., 2020), the strategy trajectories $\{(\mu_h^k, \nu_h^k)\}_{h=1,k=1}^{H,K}$ themselves are not guaranteed to converge to an approximate NE. Instead, based on $\{(\mu_h^k, \nu_h^k)\}_{h=1,k=1}^{H,K}$, we extract an auxiliary set of policies $\{(\bar{\mu}_h^k, \bar{\nu}_h^k)\}_{h=1,k=1}^{H,K}$ that leads to an approximate NE. Our construction of the auxiliary policies, largely inspired by the "certified policies" (Bai et al., 2020) for learning in two-player zero-sum games, is formally presented in Algorithm 3. At the step $h$ and state $s_h$, $\bar{\mu}_h^k$ first executes the strategy $\mu_h^k(\cdot \mid s_h)$. It then uniformly samples an episode index $k'$ from the total $\check{N}_h^k(s_h)$ episodes that the state $s_h$ was visited during the last stage. Finally, $\bar{\mu}_h^k$ repeats a similar process at step $h+1$ and episode $k'$, and so on. We define $\bar{\nu}_h^k$ analogously. Compared with the certified policies in zero-sum games, our team problem has the additional challenge that the NE value is not necessarily unique, and we need to first specify which NE to compare with before we are able to define the notion of regret. Therefore, the analytical method of bounding the duality gap in zero-sum games (Bai et al., 2020) does not apply to our problem. We also remark that our construction of the auxiliary policies slightly simplifies the "certified policies" in the sense that we only need to uniformly sample an episode index at each step, while the certified policies rely on a rather complicated weighted sampling. Such simplification is a natural benefit of our stage-based V-learning that assigns uniform weights to all time steps of the same stage in history.

Clearly, the policies $(\bar{\mu}_h^k, \bar{\nu}_h^k)$ are non-Markov, and many notations defined for Markov policies in Section 2 no longer hold. Nonetheless, we can still define the value function and the $Q$-function starting from a step $h \in [H]$ for a pair of non-Markov policies $(\bar{\mu}_h^k, \bar{\nu}_h^k)$ in a similar way as in (1), because $(\bar{\mu}_h^k, \bar{\nu}_h^k)$ does not depend on the history before the $h$-th step. For notational convenience, we introduce the operators $\mathbb{P}_h V(s, a, b) = \mathbb{E}_{s' \sim P_h(\cdot \mid s, a, b)} V(s')$ for any value function $V$, and $\mathbb{D}_{\mu \times \nu} Q(s) = \mathbb{E}_{(a,b) \sim (\mu, \nu)} Q(s, a, b)$. With these notations, the Bellman equations can be rewritten more succinctly as $Q_h^{\mu, \nu}(s, a, b) = \left( r_h + \mathbb{P}_h V_{h+1}^{\mu, \nu} \right)(s, a, b)$, and $V_h^{\mu, \nu}(s) = \left( \mathbb{D}_{\mu_h \times \nu_h} Q_h^{\mu, \nu} \right)(s)$ for any $(s, a, b, h) \in \mathcal{S} \times \mathcal{A} \times \mathcal{B} \times [H]$. For each $h \in [H], s \in \mathcal{S}$, define $V_{H+1}^{\star, \bar{\nu}_{H+1}^k}(s) = 0$, and

$$V_h^{\star, \bar{\nu}_h^k}(s) \overset{\text{def}}{=} \frac{1}{\check{n}} \sum_{i=1}^{\check{n}} \max_{\mu} \mathbb{D}_{\mu \times \nu_h^{\check{l}_i}} \left( r_h + \mathbb{P}_h V_{h+1}^{\star, \bar{\nu}_{h+1}^{\check{l}_i}} \right)(s), \tag{3}$$

where the stages are partitioned in the same way as in the execution of Algorithm 2. In particular, if the visitation to a certain state $s$ is in its first stage, we let $V_h^{\star, \bar{\nu}_h^k}(s) = H - h + 1$ instead. We emphasize

that the maximum in (3) is taken with respect to each individual episode $\check{l}_i$, and hence $V_h^{\star, \bar{\nu}_h^k}(s)$ can be considered as the value of the "dynamic best response" with respect to the non-Markov policy $\bar{\nu}_h^k$. Such a definition is stronger than the usual concept of "best fixed response in hindsight", and is crucial to ensure that we obtain a NE in the end, instead of only a coarse correlated equilibrium (see also Remark 1 for detailed discussions). Define $V_h^{\bar{\mu}_h^k, \star}(s)$ analogously.

In the following, we start with an intermediate result, which states that the optimistic value function $\overline{V}_h^k(s)$ is an upper bound of both $V_h^{\star, \bar{\nu}_h^k}(s)$ and $V_h^{\bar{\mu}_h^k, \star}(s)$ in expectation.

**Lemma 1.** $\mathbb{E}[\overline{V}_h^k(s)] \geq \mathbb{E}[V_h^{\star, \bar{\nu}_h^k}(s)]$ and $\mathbb{E}[\overline{V}_h^k(s)] \geq \mathbb{E}[V_h^{\bar{\mu}_h^k, \star}(s)], \forall (s, h, k) \in \mathcal{S} \times [H] \times [K]$.

The proof of Lemma 1 relies on our results from Section 3. Specifically, we can show that within each (stationary) stage, the agents face a matrix team problem that lasts for $\check{n}$ rounds, whose reward function is associated with the optimistic value function $\overline{V}_h^k(s)$ at the given state $s$. Since in Algorithm 2, both agents essentially run an individual copy of Algorithm 1 for each state, our analysis exactly reduces to the investigation of Algorithm 1 in Section 3. We also remark that optimistic Q-learning with the celebrated learning rate of $\alpha_t = \frac{H+1}{H+t}$ (Jin et al., 2018) could fail at this step, because it induces a matrix team problem with weighted rewards where the weights vary over time and cannot be pre-computed. Such time-varying rewards are not compatible with our SGD framework, and as far as we know it is a challenging problem in stochastic non-convex optimization on its own. Interested readers are referred to Remarks 2 and 3 in Appendix G for detailed discussions of this technical challenge. We bypassed such a challenge by utilizing stage-based V-learning instead, which in essence assigns invariant and pre-computable weights to each time step in history. Our results hence advocate the use of *stage-based* V-learning in MARL, over the seminal work Jin et al. (2018), because the former creates a stage-wise stationary environment, and by nature provides a solution to the core issue of non-stationarity (Busoniu et al., 2008; Zhang et al., 2021a) in MARL.

Suppose that for every $(h, k) \in [H] \times [K]$, $\bar{\mu}_h^k$ and $\bar{\nu}_h^k$ use a common random seed to sample the episode index from $\{1, \ldots, \check{N}_h^k(s)\}$. The value function for $(\bar{\mu}_h^k, \bar{\nu}_h^k)$ can be written recursively as

$$V_h^{\bar{\mu}_h^k, \bar{\nu}_h^k}(s) = \frac{1}{\check{n}} \sum_{i=1}^{\check{n}} \mathbb{D}_{\mu_h^{\check{l}_i} \times \nu_h^{\check{l}_i}} \left( r_h + \mathbb{P}_h V_{h+1}^{\bar{\mu}_{h+1}^{\check{l}_i}, \bar{\nu}_{h+1}^{\check{l}_i}} \right)(s),$$

and $V_{H+1}^{\bar{\mu}_{H+1}^k, \bar{\nu}_{H+1}^k}(s) = 0$. The following result shows that, in expectation (over the randomness of the reward sequence and the agents' selected actions), the agents have no incentive to deviate from the policy pairs $\{(\bar{\mu}_1^k, \bar{\nu}_1^k)\}_{k=1}^K$, up to an error term of the order $O(K^{-\frac{1}{8}})$.

**Theorem 3.** The auxiliary policies $\{(\bar{\mu}_h^k, \bar{\nu}_h^k)\}_{h=1, k=1}^{H, K}$ satisfy that

$$\frac{1}{K} \sum_{k=1}^K \mathbb{E} \left[ V_1^{\star, \bar{\nu}_1^k}(s_1) - V_1^{\bar{\mu}_1^k, \bar{\nu}_1^k}(s_1) \right] \leq O(H^{\frac{17}{8}} S^{\frac{1}{8}} A_{\max}^{\frac{3}{2}} / K^{\frac{1}{8}}), \text{ and}$$

$$\frac{1}{K} \sum_{k=1}^K \mathbb{E} \left[ V_1^{\bar{\mu}_1^k, \star}(s_1) - V_1^{\bar{\mu}_1^k, \bar{\nu}_1^k}(s_1) \right] \leq O(H^{\frac{17}{8}} S^{\frac{1}{8}} A_{\max}^{\frac{3}{2}} / K^{\frac{1}{8}}).$$

An immediate corollary is that, if we let the agents uniformly sample an episode index $\kappa$ from $\{1, \ldots, K\}$, the policy pair $(\bar{\mu}_1^\kappa, \bar{\nu}_1^\kappa)$ we constructed is an approximate NE.

**Theorem 4.** Suppose that the two agents run Algorithm 2 for $K$ episodes with $K = \Omega(1/\varepsilon^8) \cdot poly(H, S, A_{\max})$, and uniformly sample an episode index $\kappa$ from $\{1, \ldots, K\}$. Then, with probability at least $\frac{3}{5}$, the policy pair $(\bar{\mu}_1^\kappa, \bar{\nu}_1^\kappa)$ is an $\varepsilon$-approximate Nash equilibrium in expectation. A standard boosting technique (Mitzenmacher & Upfal, 2017) can increase the success probability to an arbitrary value $1 - p$ for $p \in (0, \frac{2}{5})$, by repeating the process for $O(\log \frac{1}{p})$ times.

**Remark 1.** Theorem 4 states that we obtain an approximate NE with high probability for any realization of $\kappa$, rather than only in expectation over the randomness of $\kappa$. The former condition guarantees that each outcome $(\bar{\mu}_1^\kappa, \bar{\nu}_1^\kappa)$ of the sampling process is a Nash equilibrium with high probability; with the latter condition, we can only conclude that the uniform distribution over $\{(\bar{\mu}_1^k, \bar{\nu}_1^k)\}_{k=1}^K$ is a coarse correlated equilibrium, a weaker solution concept, where the common random seed plays the role of the "trusted coordinator" in the language of correlated equilibria.

Our sample complexity bound depends on $\max\{A, B\}$ instead of $A \times B$. This is the benefit from decentralized learning, and would not have been achieved by centralized approaches. Such an improvement would become more significant as the number of agents $N$ increases. Our decentralized approach only depends on the largest single action space $\max_{i\in\mathcal{N}} |\mathcal{A}^i|$, while the centralized methods would have an exponential dependence $\Pi_{i=1}^N |\mathcal{A}^i|$. Also note that our sample complexity bound in Theorem 4 holds in expectation. To obtain a standard high-probability result that holds with probability $1 - p$, one could either immediately apply Markov's inequality and tolerate an additional $O(1/p)$ factor of computation, or replace our SGD subroutine with one that has high-probability guarantees instead (Li & Orabona, 2020). We leave such improvements to our future work as they diverge from the main focus of this paper. Similar to Section 3, the sample complexity in Theorem 4 can be further improved to $\widetilde{O}(1/\varepsilon^4)$ by incorporating variance reduction, which is sketched in Theorem 7 of Appendix G. We remark that we have not attempted to optimize the dependence on the other parameters $H, S$ and $A_{\max}$. There are many straightforward ways to obtain tighter bounds in these parameters, such as making the hyper-parameters $\alpha_{\tilde{n}}$ and $\gamma$ dependent on $A_{\max}$. For completeness, a sample complexity lower bound in the order of $\Omega(1/\varepsilon^2) \cdot \text{poly}(H, S, A_{\max})$ is presented in Appendix G, which is achieved by a reduction to a single-agent RL problem. We leave the tightening of both the upper and lower bounds to our future work.

Finally, we show that our algorithm can nearly find the team-optimal NE in an important subclass of Markov teams named *smooth teams*. Our definition of a $(\lambda, \rho)$-smooth team, adapted from the definition of smooth games (Roughgarden, 2009; Radanovic et al., 2019), is formally introduced in Definition 5 of Appendix G. Define $V^\star$ to be the team-optimal value function, i.e., $V_h^\star(s) = \max_\pi V_h^\pi(s)$ for any $h \in [H], s \in \mathcal{S}$. The following theorem states that the output policies of Algorithm 2 converge to a $\lambda/(1 + \rho)$ factor of team-optimality at a rate of $O(K^{-1/8})$.

**Theorem 5.** *Let $K = \Omega(1/\varepsilon^8) \cdot poly(H, S, A_{\max})$. In a $(\lambda, \rho)$-smooth team, the value of the auxiliary policies $\{(\bar{\mu}_h^k, \bar{\nu}_h^k)\}_{h=1,k=1}^{H,K}$ satisfies*

$$\frac{1}{K}\sum_{k=1}^K \mathbb{E}\left[V_1^{\bar{\mu}_1^k, \bar{\nu}_1^k}(s_1)\right] \geq \frac{\lambda}{1 + \rho}V_1^\star(s_1) - \frac{1}{1 + \rho}O(K^{-\frac{1}{8}}).$$

## 6 SIMULATIONS

We empirically evaluate `SGD-IX` on a classic matrix team task, and `V-learning SGD` on a Markov team. Figure 1 illustrates the performances of the algorithms in terms of both the rewards and the $L^2$ equilibrium gaps (which measure the $L^2$ distance to a NE, formally defined in (39) of Appendix H). Our simulation results turn out to be more encouraging that what our theory suggests: Both the actual policy trajectories ("Current") and the auxiliary policies ("Average") converge to NE in the tasks we tested. Further, the rewards obtained approach those of a team-optimal oracle ("Centralized"), which suggests that our algorithm achieves the team-optimal NE very frequently in our simulations, even though our theory does not guarantee so in general. Detailed descriptions of the simulations are deferred to Appendix H due to space limitations.

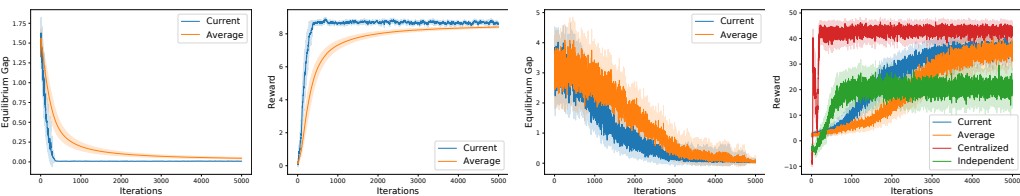

(a) Equilibrium gap (matrix)   (b) Reward (matrix)   (c) Equilibrium gap (Markov)   (d) Reward (Markov)

Figure 1: (a) $L^2$ equilibrium gaps and (b) rewards of Algorithm 1 on the matrix team task, and (c) $L^2$ equilibrium gaps and (d) rewards of Algorithm 2 on the Markov team task. "Current" denotes the actual policy trajectory $\{(\mu_h^k, \nu_h^k)\}_{h=1,k=1}^{H,K}$, while "Average" represents the auxiliary policies $\{(\bar{\mu}_h^k, \bar{\nu}_h^k)\}_{h=1,k=1}^{H,K}$. In "Independent", each agent runs a naïve single-agent Q-learning algorithm independently, by taking greedy actions w.r.t its local Q-function estimates. Shaded areas denote the standard deviations of the equilibrium gap or reward.

ETHICS STATEMENT

As a theory-oriented work, we do not believe that our research will cause any ethical issue, or put anyone at any disadvantage. In particular, we do not believe that our work will cause any discrimination / bias / fairness concerns, privacy and security issues, or legal compliance, and so on.

REPRODUCIBILITY STATEMENT

For every theoretical claim in the paper, we included a clear statement and explanation of all necessary assumptions right before or after the theoretical claim. The complete proofs of all theoretical results in the paper can be found in Appendices E, F, and G. The source code used in the simulations are uploaded as supplementary materials along with the submission of the paper.

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

# Supplementary Materials for "Decentralized Cooperative Multi-Agent Reinforcement Learning with Exploration"

## A  DETAILED DISCUSSIONS ON RELATED WORK

A common mathematical framework of multi-agent RL is stochastic games (Shapley, 1953), which is also referred to as Markov games. Early attempts to learn Nash equilibria in Markov games include Littman (1994; 2001); Hu & Wellman (2003); Hansen et al. (2013), but they either assume the transition kernel and rewards are known, or only yield asymptotic guarantees. More recently, various sample efficient methods have been proposed (Wei et al., 2017; Bai & Jin, 2020; Sidford et al., 2020; Xie et al., 2020; Bai et al., 2020; Liu et al., 2021; Zhao et al., 2021), mostly for learning in two-player zero-sum Markov games. Most notably, several works have investigated two-player zero-sum games in a *decentralized* environment: Daskalakis et al. (2020) have shown non-asymptotic convergence guarantees for independent policy gradient methods when the learning rates of the two agents follow a two-timescale rule. Tian et al. (2021) have studied online learning when the actions of the opponents are not observable, and have achieved the first sub-linear regret $\widetilde{O}(K^{\frac{3}{4}})$ in the decentralized setting for $K$ episodes. More recently, Wei et al. (2021) have proposed an Optimistic Gradient Descent Ascent algorithm with a slowly-learning critic, and have shown a strong finite-time last-iterate convergence result in the decentralized/agnostic environment. Overall, these works have mainly focused on two-player zero-sum games. These results do not carry over in any way to the decentralized cooperative/team setting that we consider in this paper.

Another line of research has considered RL in teams or cooperative games. Without enforcing a decentralized environment, Boutilier (1996) has proposed to coordinate the agents by letting them take actions in a lexicographic order. In a similar setting, Wang & Sandholm (2002) have studied optimal adaptive learning that converges to the optimal NE in Markov teams. Verbeeck et al. (2002) have presented an independent learning algorithm that achieves a Pareto optimal Nash equilibrium in common interest games with limited communication. These methods critically relied on communications among the agents (beforehand) or observing the teammates' actions. In contrast, the distributed Q-learning algorithm in Lauer & Riedmiller (2000) is decentralized and coordination-free, which, however, only works for deterministic tasks, and has no non-asymptotic guarantees. In fact, developing provable decentralized RL in teams with stochastic environment is considered as an open problem in Lauer & Riedmiller (2000), which we address in this paper.

Another relevant work (Arslan & Yüksel, 2016) has shown that decentralized Q-learning can converge to Nash equilibria in weakly acyclic games (WAGs), which cover Markov teams as an important special case. Their decentralized setting is most similar to ours in that each agent is completely oblivious to the presence of the others. Later, such a method has been improved in Yongacoglu et al. (2019) to achieve team-optimality. However, both of them require a coordinated exploration phase, and only yield asymptotic guarantees. In fact, to the best of our knowledge, it is not clear if their methods will converge in polynomial time. We significantly improve their results in this respect, by explicitly showing a polynomial sample complexity upper bound. Similarly, decentralized learning has also been studied in single-stage weakly acyclic games (Marden et al., 2009b) or potential games (a subclass of WAGs) (Marden et al., 2009a; Cohen et al., 2017). Two contemporaneous works (Zhang et al., 2021b; Leonardos et al., 2021) have studied gradient play in Markov potential games, which also cover Markov teams. Nonetheless, they do not consider the aspect of *exploring* the unknown environment, one of the key issues in RL, and/or have to assume perfect knowledge of the environment.

In general-sum normal-form games, a folklore result is that when the agents independently run no-regret learning algorithms, their empirical frequency of plays converges to the set of coarse correlated equilibria (CCE) of the game (Hart & Mas-Colell, 2000). However, a CCE may suggest that the agents play obviously non-rational strategies. For example, Viossat & Zapechelnyuk (2013) have constructed an example where a CCE assigns positive probabilities only to strictly dominated strategies. On the other hand, given the PPAD completeness of finding a Nash equilibrium, convergence to NE seems hopeless in general. An impossibility result (Hart & Mas-Colell, 2003) has shown that uncoupled no-regret learning does not converge to Nash equilibrium in general, due to the informational constraint

that the adjustment in an agent's strategy does not depend on the reward functions of the others. Hence, convergence to Nash equilibria is guaranteed mostly in games with special reward structures, such as two-player zero-sum games (Freund & Schapire, 1999) and potential games (Kleinberg et al., 2009; Cohen et al., 2017).

Efficient exploration has also been widely studied in the literature of single-agent RL, see, e.g., Brafman & Tennenholtz (2002); Jaksch et al. (2010); Azar et al. (2017); Jin et al. (2018); Agarwal et al. (2020). For the tabular episodic setting, various methods (Azar et al., 2017; Zhang et al., 2020b; Menard et al., 2021) have achieved the sample complexity of $\widetilde{O}(H^3 SA/\varepsilon^2)$, which matches the information-theoretical lower bound. When reduced to the bandit case, our problem is also related to the cooperative multi-armed bandit (MAB) problem (Lai et al., 2008; Avner & Mannor, 2014; Kalathil et al., 2014; Bubeck & Budzinski, 2020), originated from the literature of cognitive radio networks. The difference is that, in cooperative MAB, each agent is essentially interacting with an individual copy of the bandit, with an extra caution of action collisions; in our formulation, the reward function is defined on the Cartesian product of the action spaces, which allows the agents to be coupled in more general forms. A concurrent work (Chang et al., 2021) has studied cooperative multi-player multi-armed bandits with information asymmetry, which parallels our matrix team setting. Nevertheless, (Chang et al., 2021) requires stronger conditions than our decentralized setting as their algorithm relies on playing a predetermined sequence of actions.

## B    DISCUSSIONS ON MARKOV POTENTIAL GAMES

An $N$-player episodic Markov potential game can be described by a tuple $\mathcal{G} = (\mathcal{N}, H, \mathcal{S}, \{\mathcal{A}^i\}_{i=1}^N, \{r^i\}_{i=1}^N, P)$, where (1) $\mathcal{N} = \{1, 2, \ldots, N\}$ is the set of agents; (2) $H \in \mathbb{N}_+$ is the number of time steps in each episode; (3) $\mathcal{S}$ is the finite state space; (4) $\mathcal{A}^i$ is the finite action space for agent $i \in \mathcal{N}$; (5) $r^i : [H] \times \mathcal{S} \times \mathcal{A} \to [0, 1]$ is the reward function for agent $i$, where $\mathcal{A} = \times_{i=1}^N \mathcal{A}^i$ is the joint action space; and (6) $P : [H] \times \mathcal{S} \times \mathcal{A} \to \Delta(\mathcal{S})$ is the transition kernel. We remark that the agents could have different reward functions, but the reward that an agent receives also depends on the joint actions of all the agents.

We define the space of policies for agent $i$ by $\Pi^i = \{\pi^i : [H] \times \mathcal{S} \to \Delta(\mathcal{A}^i)\}$. The joint policy space is denoted by $\Pi = \times_{i=1}^N \Pi^i$. For a joint policy $\pi \in \Pi$, and for any $h \in [H]$, $s \in \mathcal{S}$, and $a \in \mathcal{A}$, we define the value function for agent $i$ as:

$$V_h^{\pi,i}(s) \stackrel{\text{def}}{=} \mathbb{E}_{s_{h'+1} \sim P_h(\cdot|s_{h'}, a_{h'})} \left[ \sum_{h'=h}^H r_{h'}^i(s_{h'}, a_{h'}) \mid s_h = s \right]. \tag{4}$$

We call $\mathcal{G}$ a Markov potential game if for every $h \in [H]$ and $s \in \mathcal{S}$, there exists a function $\Phi_h^s : \Pi \to \mathbb{R}$, such that

$$\Phi_h^s(\pi^i, \pi^{-i}) - \Phi_h^s(\pi^{i'}, \pi^{-i}) = V_h^{(\pi^i, \pi^{-i}), i}(s) - V_h^{(\pi^{i'}, \pi^{-i}), i}(s),$$

for all $i \in \mathcal{N}, \pi^i, \pi^{i'} \in \Pi^i$ and $\pi^{-i} \in \Pi^{-i}$. Such a definition extends the normal-form potential games (Monderer & Shapley, 1996) to state-dependent games. In words, $\Phi_h^s$ can be considered as a potential function such that any unilateral change in an agent's policy causes equal amount of change to the potential function and its own value function. It is known that every Markov potential game possesses at least one deterministic Nash equilibrium.

We can easily see that Markov teams are a special case of Markov potential games, where the potential function is simply the team value function. Since all the agents are correlated by a single potential function in Markov potential games, the process of learning Nash equilibria reduces to searching for the local optima of the potential function, which is exactly the core idea behind our design of Algorithm 1 for Markov teams. One can see that Lemmas 4 and 5 immediately generalize to Markov potential games, and hence the same sample complexity bounds from Theorems 1, 2 and 4 also hold in such a broader class of games.

## C    AUXILIARY LEMMAS

### C.1    PROPERTIES OF THE BREGMAN DIVERGENCE

Let $\omega : \mathcal{X} \to \mathbb{R}$ be a continuously differentiable mirror map that is 1-strongly convex on $\mathcal{X}$, where $\mathcal{X}$ is a non-empty closed convex set. In this paper, we specifically take $\omega$ as the $L^2$-induced regularizer, i.e., $\omega(x) = \frac{1}{2} \|x\|_2^2$, although the following result holds more generally. Let $D_\omega(x, y) = \omega(x) - \omega(y) - \langle x - y, \nabla\omega(y) \rangle$ be the $\omega$-induced Bregman divergence.

**Lemma 2.** *(Beck, 2017, Section 9.2.1). The Bregman divergence satisfies the following properties:*

*(a) Three-point identity:*

$$D_\omega(x, y) + D_\omega(y, z) - D_\omega(x, z) = \langle \nabla\omega(z) - \nabla\omega(y), x - y \rangle, \forall x, y, z \in \mathcal{X}.$$

*(b) Suppose $\varphi(x)$ is convex and $z^+ = \underset{x \in \mathcal{X}}{\arg\min} \left\{ \varphi(x) + \frac{1}{\alpha} D_\omega(x, z) \right\}$ for some $\alpha > 0$, we have*

$$\varphi(x) + \frac{1}{\alpha} D_\omega(x, z) \geq \varphi\left(z^+\right) + \frac{1}{\alpha} D_\omega\left(z^+, z\right) + \frac{1}{\alpha} D_\omega\left(x, z^+\right), \forall x \in \mathcal{X}.$$

### C.2    PROPERTIES OF THE GRADIENT MAPPING

**Lemma 3.** *(Agarwal et al., 2021, Proposition B.1). Let $L : \mathcal{X} \to \mathbb{R}$ be a $\beta$-smooth function. Define the gradient mapping as*

$$G^\eta(x) = \frac{1}{\eta} \left( \Pi_\mathcal{X} \left( x + \eta\nabla L(x) \right) - x \right).$$

*The update rule for projected gradient ascent is $x^+ = x + \eta G^\eta(x)$. If $\|G^\eta(x)\|_2 \leq \varepsilon$, then*

$$\max_{x + \delta \in \mathcal{X}, \|\delta\|_2^2 \leq 1} \delta^\intercal \nabla L(x^+) \leq \varepsilon(\eta\beta + 1).$$

## D    AN EXTENDED DISCUSSION OF SECTION 3

We first introduce a few more notations for ease of presentation. Recall that $r : \mathcal{A} \times \mathcal{B} \to [0, 1]$ is the expected value of the reward function, i.e., $r(a, b) = \int_0^1 x dP_{a,b}(x)$. We further let $Q \in \mathbb{R}^A \times \mathbb{R}^B$ be the matrix of expected rewards. That is, given a mixed strategy pair $\pi = (\mu, \nu)$, the corresponding expected reward for the players can be denoted by $V(\pi) = V(\mu, \nu) = \mathbb{E}_{\mu \times \nu}[r(a, b)] = \mu^\intercal Q\nu$, where $a \sim \mu$ and $b \sim \nu$. The function $V$ is also called the value function.

In certain circumstances, it is more convenient to work with costs instead of rewards. We let $l : \mathcal{A} \times \mathcal{B} \to [0, 1]$ be the expected cost function such that $l(a, b) = 1 - r(a, b), \forall (a, b) \in \mathcal{A} \times \mathcal{B}$. Let $M = \mathbf{1}_{A \times B} - Q$ be the matrix of expected costs, where $\mathbf{1}_{A \times B}$ is an all-one matrix of the size $A \times B$. Similar to the value function $V$, we define $L(\pi) = L(\mu, \nu) \stackrel{\text{def}}{=} \mathbb{E}_{\mu \times \nu}[l(a, b)] = \mu^\intercal M\nu$ as the corresponding loss function.

Our definition of an approximate Nash equilibrium (Definition 1) can be easily reduced to the matrix team setting as follows.

**Definition 2.** *($\varepsilon$-approximate NE for matrix teams). For $\varepsilon > 0$, a strategy pair $(\mu^\star, \nu^\star)$ is an $\varepsilon$-approximate Nash equilibrium for a matrix team if $V(\mu^\star, \nu^\star) \geq V(\mu, \nu^\star) - \varepsilon, \forall \mu \in \Delta(\mathcal{A})$, and $V(\mu^\star, \nu^\star) \geq V(\mu^\star, \nu) - \varepsilon, \forall \nu \in \Delta(\mathcal{B})$.*

In the following, we introduce the definition of an approximate stationary point.

**Definition 3.** *($\varepsilon$-approximate stationary point). A strategy pair $\pi = (\mu, \nu) \in \Delta(\mathcal{A}) \times \Delta(\mathcal{B})$ is a (first-order) $\varepsilon$-approximate stationary point for the function $V : \Delta(\mathcal{A}) \times \Delta(\mathcal{B}) \to \mathbb{R}$ if for any $\delta_1 \in \mathbb{R}^A, \delta_2 \in \mathbb{R}^B$ such that $\|\delta_1\|_2^2 + \|\delta_2\|_2^2 \leq 1, \mu + \delta_1 \in \Delta(\mathcal{A})$, and $\nu + \delta_2 \in \Delta(\mathcal{B})$, it holds that*

$$\delta_1^\intercal \nabla_\mu V(\pi) + \delta_2^\intercal \nabla_\nu V(\pi) \leq \varepsilon.$$

Intuitively, $\pi$ is a stationary point if the value function $V$ cannot increase by more than $\varepsilon$ along any direction that lies in the intersection of the strategy simplex and the neighborhood of $\pi$. Comparing the definitions of an approximate stationary point with an approximate NE easily leads to the following observation.

**Lemma 4.** *Let $\pi^\star = (\mu^\star, \nu^\star)$ be an $\varepsilon$-approximate stationary point of the value function $V$ for some $\varepsilon > 0$. Then, the strategy pair $(\mu^\star, \nu^\star)$ is a $\sqrt{2}\varepsilon$-approximate Nash equilibrium of the corresponding matrix team.*

All missing proofs of this section are deferred to Appendix E for clarity. With Lemma 4, learning a NE of the team reduces to finding a stationary point of the corresponding value function, or equivalently, the loss function. In what follows, we focus on the loss function $L$ for reasons that will be clear soon. Numerous SGD methods have been proposed to find a stationary point in stochastic non-convex optimization (Ghadimi & Lan, 2013). For generality, we write our gradient step as an Online Mirror Descent (OMD) update with $L^2$ regularization to search for the stationary point. The standard OMD algorithm performs the following update rule at each iteration:

$$x_{t+1} \leftarrow \operatorname*{argmin}_{x \in \mathcal{X}} \left\{ \langle \widehat{\nabla} L(x_t), x \rangle + \frac{1}{\alpha_t} D_\omega(x, x_t) \right\}, \tag{5}$$

where $\mathcal{X} = \Delta(\mathcal{A}) \times \Delta(\mathcal{B})$ is the feasible region, $\widehat{\nabla} L(x_t)$ is an estimator of the subgradient of $L$ at $x_t$, $\alpha_t > 0$ is the stepsize, $\omega : \mathcal{X} \to \mathbb{R}$ is a 1-strongly convex and continuously differentiable regularizer, and $D_\omega(x, x_t) = \omega(x) - \omega(x_t) - \langle \nabla \omega(x_t), x - x_t \rangle$ is the Bregman divergence induced by the regularizer $\omega$.

The following nuance in the general OMD update requires some attention before we directly apply existing results. Since the optimization objective $L$ is a function of $\pi \in \Delta(\mathcal{A}) \times \Delta(\mathcal{B})$, existing methods typically proceed in a centralized way and manipulate $\pi$ as a whole. In contrast, in our decentralized setting, the strategies $\mu$ and $\nu$ need to be updated separately and independently. However, such a nuance does not exist with SGD. Specifically, if we instantiate $\omega$ as an $L^2$-induced regularizer, i.e., $\omega(x) = \frac{1}{2} \|x\|_2^2$, then our OMD procedure is equivalent to projected SGD (Lattimore & Szepesvári, 2020, Example 28.2). In this case, the Bregman divergence simplifies to $D_\omega(x, x_t) = \frac{1}{2} \|x - x_t\|_2^2$. The update rule in (5) can hence be simplified to

$$x_{t+1} \leftarrow \Pi_{\mathcal{X}} \left( x - \alpha_t \widehat{\nabla} L(x_t) \right),$$

where $\Pi_{\mathcal{X}}(x)$ denotes the Euclidean projection of $x$ onto $\mathcal{X}$. We then point out the following simple fact, which states that conducting projected SGD on the optimization variable $\pi = (\mu, \nu)$ as a whole is equivalent to running the same algorithm on each component of $\pi$ independently.

**Lemma 5.** *Let $\pi = (\mu, \nu)$ be a strategy pair, and let $\widehat{\nabla}_\pi L(\pi) = (\widehat{\nabla}_\mu L(\pi), \widehat{\nabla}_\nu L(\pi))$ be an arbitrary (not necessarily unbiased) gradient estimator of $L$ at $\pi$. Let $\pi^+ = \Pi_{\Delta(\mathcal{A}) \times \Delta(\mathcal{B})}(\pi - \alpha \widehat{\nabla}_\pi L(\pi))$ be the result of a centralized gradient update step with stepsize $\alpha > 0$, and let $\mu^+ = \Pi_{\Delta(\mathcal{A})}(\mu - \alpha \widehat{\nabla}_\mu L(\pi))$ and $\nu^+ = \Pi_{\Delta(\mathcal{B})}(\nu - \alpha \widehat{\nabla}_\nu L(\pi))$ be the results of applying the gradient update to each agent's strategy independently. Then, $\pi^+ = (\mu^+, \nu^+)$.*

Therefore, it suffices to only analyze the centralized OMD algorithm with $L^2$ regularization, as it is equivalent to decentralized SGD. Since our algorithm is essentially performing gradient descent instead of ascent, we choose the loss function $L$ as our optimization objective.

We now turn to the gradient estimator $\widehat{\nabla} L(x)$. Existing stochastic non-convex optimization methods (Ghadimi & Lan, 2013) typically assume the existence of a stochastic oracle: Given any input $x \in \mathcal{X}$, the oracle outputs a random vector $\widehat{\nabla} L(x, \xi)^3$ that is an unbiased estimator of the gradient and has bounded variance. More specifically, it is assumed that for any $x \in \mathcal{X}$, $\mathbb{E}_\xi[\widehat{\nabla} L(x, \xi)] \in \partial L(x)$, where $\partial L(x)$ is the subdifferential set of $L$ at $x$, and $\mathbb{E}_\xi[\|\widehat{\nabla} L(x, \xi)\|_2^2] \leq \rho^2$ for some constant $\rho > 0$. The latter condition is also called the $\rho$-stochastic continuity of $L$ (Lu, 2019). Such an oracle does not come for free in our problem, as the agents are faced with bandit feedback and can only observe

---

[3]$\xi$ is a random variable that is used to explicitly show the randomness of the estimator. When there is no possibility of ambiguity, we suppress the $\xi$ term and simply write the estimator as $\widehat{\nabla} L(x)$.

the reward associated with the selected action pair. To deal with this challenge, at the $t$-th round of the problem, we let agent 1 construct a biased gradient estimator $\widetilde{\nabla}_\mu L_t$ in the form of

$$\widetilde{\nabla}_{\mu,i} L_t = \frac{1 - R_t}{\mu_{t,i} + \gamma} \mathbb{I}\{a_t = i\}, \forall i \in \mathcal{A},$$

where $\mu_{t,i}$ is the probability that agent 1 takes action $i \in \mathcal{A}$ in round $t$, and $\gamma > 0$ is a constant specified in Algorithm 1. The biased gradient estimator $\widetilde{\nabla}_\nu L_t$ for agent 2 is constructed similarly. We can see that the estimator we use tends to underestimate the actual cost. Such a technique is called Implicit eXploration (Neu, 2015) in the literature, and is widely used to prove high-probability regret bounds in multi-armed bandits, an objective that we do not pursue in this paper.

Our analysis of Algorithm 1 relies on showing that the biased estimator $\widetilde{\nabla} L_t$ we constructed has bounded variance, and is upper bounded by the actual gradient in expectation, which satisfies the oracle requirements in a weaker sense. We can then analyze the non-asymptotic convergence behavior of our SGD algorithm to a stationary point. For generality, the proof we present in this paper follows the analysis of a general proximal point method (Davis & Drusvyatskiy, 2018; Zhang & He, 2018), by showing that the gradient of the Moreau envelope converges to 0, although simpler proofs for the convergence of SGD in non-convex optimization exist. Finally, we invoke Lemma 4 to conclude that the stationary point we find is close to a NE.

# E  MISSING RESULTS IN SECTION 3 AND APPENDIX D

## E.1  PROOF OF LEMMA 4

*Proof.* Let $\pi^\star = (\mu^\star, \nu^\star)$ be an $\varepsilon$-approximate stationary point of the value function $V$. For any strategy $\mu \in \Delta(\mathcal{A})$,

$$V(\mu, \nu^\star) - V(\mu^\star, \nu^\star) = (\mu - \mu^\star)^\intercal Q \nu^\star = \sqrt{2} \cdot \frac{1}{\sqrt{2}} (\mu - \mu^\star)^\intercal \nabla_\mu V(\pi^\star) \le \sqrt{2}\varepsilon,$$

where the last step follows from the $\varepsilon$-approximate stationarity of $\pi^\star$ and $\left\| \frac{1}{\sqrt{2}} (\mu - \mu^\star) \right\|_2^2 \le 1$. A similar argument can show that $V(\mu^\star, \nu) - V(\mu^\star, \nu^\star) \le \sqrt{2}\varepsilon, \forall \nu \in \Delta(\mathcal{B})$. Therefore, $(\mu^\star, \nu^\star)$ is a $\sqrt{2}\varepsilon$-approximate NE of the matrix team. □

## E.2  PROOF OF LEMMA 5

*Proof.* Let $\pi = (\mu, \nu)$. First, notice that $\pi - \alpha \widehat{\nabla}_\pi L(\pi) = (\mu - \alpha \widehat{\nabla}_\mu L(\pi), \nu - \alpha \widehat{\nabla}_\nu L(\pi))$. Then, by the definition of the Euclidean projection,

$$
\begin{aligned}
\pi^+ &= \Pi_{\Delta(\mathcal{A}) \times \Delta(\mathcal{B})} \left( \pi - \alpha \widehat{\nabla}_\pi L(\pi) \right) \\
&= \underset{x \in \Delta(\mathcal{A}) \times \Delta(\mathcal{B})}{\operatorname{argmin}} \left\| \pi - \alpha \widehat{\nabla}_\pi L(\pi) - x \right\|_2^2 \\
&= \underset{x_1 \in \Delta(\mathcal{A}), x_2 \in \Delta(\mathcal{B})}{\operatorname{argmin}} \left( \left\| \mu - \alpha \widehat{\nabla}_\mu L(\pi) - x_1 \right\|_2^2 + \left\| \nu - \alpha \widehat{\nabla}_\nu L(\pi) - x_2 \right\|_2^2 \right) \\
&= \underset{x_1 \in \Delta(\mathcal{A})}{\operatorname{argmin}} \left\| \mu - \alpha \widehat{\nabla}_\mu L(\pi) - x_1 \right\|_2^2 + \underset{x_2 \in \Delta(\mathcal{B})}{\operatorname{argmin}} \left\| \nu - \alpha \widehat{\nabla}_\nu L(\pi) - x_2 \right\|_2^2 \\
&= \Pi_{\Delta(\mathcal{A})} \left( \mu - \alpha \widehat{\nabla}_\mu L(\pi) \right) + \Pi_{\Delta(\mathcal{B})} \left( \nu - \alpha \widehat{\nabla}_\nu L(\pi) \right) \\
&= \mu^+ + \nu^+.
\end{aligned}
$$

This completes the proof of the lemma. □

## E.3  PROOF OF LEMMA 7

*Proof.* Our proof relies on the smoothness of the function $L$ and a gradient mapping property (Proposition B.1 in Agarwal et al. (2021), also reproduced as Lemma 3 in our Appendix C for

completeness). First, recall the definition that $\hat{x} = \text{argmin}_{z \in \mathcal{X}} \{L(z) + \lambda D_\omega(z, x)\}$, and notice that the function $L(z) + \lambda D_\omega(z, x)$ is $(\lambda - 1)$-strongly convex in $z$ for any fixed $x \in \mathcal{X}$. We know from Lemma 2 that

$$L(x) - [L(\hat{x}) + \lambda D_\omega(\hat{x}, x)] \geq (\lambda - 1) D_\omega(x, \hat{x}).$$

Therefore, we can derive that

$$
\begin{aligned}
L(x) - L(\hat{x}) - D_\omega(\hat{x}, x) =& L(x) - (L(\hat{x}) + \lambda D_\omega(\hat{x}, x)) + (\lambda - 1) D_\omega(\hat{x}, x) \\
\geq& (\lambda - 1)(D_\omega(x, \hat{x}) + D_\omega(\hat{x}, x)) \\
=& (\lambda - 1) \|x - \hat{x}\|_2^2.
\end{aligned}
$$

It is easy to verify that $L(x)$ is an $(A + B)$-smooth function, in the sense that $\|\nabla L(x) - \nabla L(y)\|_2 \leq (A + B) \|x - y\|_2$. We can hence invoke Lemma 3 and conclude that $x$ is an approximate stationary point, such that

$$\max_{x + \delta \in \mathcal{X}, \|\delta\|_2^2 \leq 1} \delta^\intercal \nabla L(x) \leq \frac{2(2A_{\max} + \lambda)\sqrt{\varepsilon}}{\sqrt{\lambda - 1}}.$$

$\square$

### E.4 PROOF OF THEOREM 1

*Proof.* From Lemma 5, we know that it suffices to analyze Algorithm 1 from a centralized perspective, because running projected gradient descent on the joint strategy $x = (\mu, \nu)$ as a whole is equivalent to updating each agent's strategy $\mu$ and $\nu$ individually. To show that $x_\tau = (\mu_\tau, \nu_\tau)$ is an approximate Nash equilibrium, we also know from Lemma 4 that the problem reduces to showing its approximate stationarity with respect to the loss function $L$.

For generality, the proof we present in this paper follows the analysis of a general proximal point method (Davis & Drusvyatskiy, 2018; Zhang & He, 2018), although many simpler proofs are given in the literature to show the convergence of SGD in non-convex optimization. To facilitate the analysis, we first introduce the following definition of the Bregman-Moreau envelope (Moreau, 1965; Drusvyatskiy & Paquette, 2019; Davis & Drusvyatskiy, 2018; Zhang & He, 2018).

**Definition 4.** *(Bregman-Moreau envelope). For $\lambda > 0$, given a function $L : \mathcal{X} \to \mathbb{R}$ and a vector $z \in \mathcal{X}$, the Bregman-Moreau envelope is defined as*

$$L_\lambda(z) \overset{\text{def}}{=} \min_{x \in \mathcal{X}} \left\{ L(x) + \frac{1}{\lambda} D_\omega(x, z) \right\}.$$

*The corresponding Bregman proximal operator is*

$$prox_{\lambda, L}(z) \overset{\text{def}}{=} \underset{x \in \mathcal{X}}{\text{argmin}} \left\{ L(x) + \frac{1}{\lambda} D_\omega(x, z) \right\}.$$

We start with the observation that $L$ is a 1-weakly convex function, in the sense that $L(x) + \omega(x)$ is convex for $\omega(x) = \frac{1}{2} \|x\|_2^2$. It then follows that for any $0 < \lambda < 1$, the proximal operator is well-defined and unique.

**Lemma 6.** *(Zhang & He, 2018, Lemma 2.2). Suppose $L$ is 1-weakly convex on $\mathcal{X}$, and let $0 < \lambda < 1$. Then, for any vector $z \in \mathcal{X}$, the function $L(x) + \frac{1}{\lambda} D_\omega(x, z)$ is $(\lambda^{-1} - 1)$-strongly convex in $x$, and the Bregman proximal operator $prox_{\lambda, L}(z)$ is uniquely defined.*

For simplicity of notations, in the following, we use $x_t = (\mu_t, \nu_t) \in \mathcal{X}$ to denote the strategy pair used by the two agents at round $t \in [T]$. For any $\lambda > 1$ and $x \in \mathcal{X}$, we let

$$\hat{x} \overset{\text{def}}{=} prox_{1/\lambda, L}(x),$$

and we know from Lemma 6 that $\hat{x}$ is uniquely defined. By the definition of the Bregman envelope, $L_{1/\lambda}(x_{t+1}) = L(\hat{x}_{t+1}) + \lambda D_\omega(\hat{x}_{t+1}, x_{t+1})$. The optimality of $\hat{x}_{t+1}$ implies that

$$L_{1/\lambda}(x_{t+1}) \leq L(\hat{x}_t) + \lambda D_\omega(\hat{x}_t, x_{t+1}). \tag{6}$$

Let $\widetilde{\nabla} L_t = (\widetilde{\nabla}_\mu L_t, \widetilde{\nabla}_\nu L_t) \in \mathbb{R}^{A+B}$ be the biased gradient estimator constructed by concatenating the two players' individual estimates in Line 5 of Algorithm 1. Further, let $\nabla L(x_t) = (\nabla_\mu L(x_t), \nabla_\nu L(x_t)) = (M\nu_t, M^\intercal \mu)$ be the actual gradient of $L$ at $x_t$, a value that we do not have access to and is only used for analytical purposes. Our proof relies on a few properties of the Bregman divergence, which are standard results from the literature and are reproduced in Appendix C for completeness. Setting $z = x_t, z^+ = x_{t+1}, x = \hat{x}_t, \alpha = \alpha_t$, and $\varphi(x) = \langle \widetilde{\nabla} L_t, x \rangle$ in Lemma 2(b) (Appendix C) leads to

$$\alpha_t \left\langle \widetilde{\nabla} L_t, \hat{x}_t - x_{t+1} \right\rangle \geq D_\omega(\hat{x}_t, x_{t+1}) + D_\omega(x_{t+1}, x_t) - D_\omega(\hat{x}_t, x_t). \tag{7}$$

Rearranging and combining (6) and (7),

$$L_{1/\lambda}(x_{t+1}) \leq L(\hat{x}_t) + \lambda D_\omega(\hat{x}_t, x_{t+1})$$
$$\leq L(\hat{x}_t) + \lambda \left( \alpha_t \left\langle \widetilde{\nabla} L_t, \hat{x}_t - x_{t+1} \right\rangle - D_\omega(x_{t+1}, x_t) + D_\omega(\hat{x}_t, x_t) \right)$$
$$= L_{1/\lambda}(x_t) + \lambda \left( \alpha_t \left\langle \widetilde{\nabla} L_t, \hat{x}_t - x_{t+1} \right\rangle - D_\omega(x_{t+1}, x_t) \right)$$
$$= L_{1/\lambda}(x_t) + \lambda \alpha_t \left\langle \widetilde{\nabla} L_t, \hat{x}_t - x_t \right\rangle + \lambda \left( \alpha_t \left\langle \widetilde{\nabla} L_t, x_t - x_{t+1} \right\rangle - D_\omega(x_{t+1}, x_t) \right), \tag{8}$$

where the first equality is by the definition of the Bregman-Moreau envelope. The second term in (8) can be further decomposed into

$$\langle \widetilde{\nabla} L_t, \hat{x}_t - x_t \rangle = \underbrace{\langle \nabla L(x_t), \hat{x}_t - x_t \rangle}_{\text{①}} + \underbrace{\langle \widetilde{\nabla} L_t - \nabla L(x_t), \hat{x}_t \rangle}_{\text{②}} - \underbrace{\langle \widetilde{\nabla} L_t - \nabla L(x_t), x_t \rangle}_{\text{③}}. \tag{9}$$

In the following, we bound (the expected value of) each term in (9) separately. First, since $L$ is 1-weakly convex on $\mathcal{X}$, the first term in (9) is bounded by

$$\text{①} = \langle \nabla L(x_t), \hat{x}_t - x_t \rangle \leq L(\hat{x}_t) - L(x_t) + \frac{1}{2} \|\hat{x}_t - x_t\|_2^2 = L(\hat{x}_t) - L(x_t) + D_\omega(\hat{x}_t, x_t). \tag{10}$$

Second, by our construction of the biased estimator, for any $i \in \mathcal{A}$,

$$\mathbb{E}[\widetilde{\nabla}_{\mu,i} L_t] = \mathbb{E}\left[ \frac{1 - R_t}{\mu_{t,i} + \gamma} \mathbb{I}\{a_t = i\} \right]$$
$$\leq \mathbb{E}\left[ \frac{1 - R_t}{\mu_{t,i}} \mathbb{I}\{a_t = i\} \right] = \sum_{j \in \mathcal{B}} \mathbb{P}(b_t = j) \cdot \mathbb{E}\left[ \frac{1 - R_t}{\mu_{t,i}} \mathbb{I}\{a_t = i\} \right]$$
$$= \sum_{j \in \mathcal{B}} \mathbb{P}(b_t = j) \cdot l(i, j) = \nabla_{\mu,i} L(x_t),$$

where here $\mathbb{E}[\cdot]$ denotes the conditional expectation given the history of both agents up to the current round, and $\nabla_{\mu,i} L(x_t)$ denotes the $i$-th entry of the actual gradient $\nabla_\mu L(x_t)$. A similar result holds for agent 2, that is, $\mathbb{E}[\widetilde{\nabla}_{\nu,j} L_t] \leq \nabla_{\nu,j} L(x_t), \forall j \in \mathcal{B}$. Hence, we know that $\mathbb{E}[\widetilde{\nabla} L_t - \nabla L(x_t)] \preceq \mathbf{0}$, where $\preceq$ denotes element-wise comparison. Since $\hat{x}_t$ lies in the the domain $\mathcal{X} = \Delta(\mathcal{A}) \times \Delta(\mathcal{B})$, it is a non-negative vector, i.e., $\hat{x}_t \succeq \mathbf{0}$. We can therefore upper bound the expected value of the second term in (9) by

$$\mathbb{E}[\text{②}] = \mathbb{E}[\langle \widetilde{\nabla} L_t - \nabla L(x_t), \hat{x}_t \rangle] \leq 0. \tag{11}$$

Finally, recalling that $x_t = (\mu_t, \nu_t)$, we decompose the third term in (9) into two parts,

$$-\text{③} = -\langle \widetilde{\nabla} L_t - \nabla L(x_t), x_t \rangle = \langle \nabla_\mu L(x_t) - \widetilde{\nabla}_\mu L_t, \mu_t \rangle + \langle \nabla_\nu L(x_t) - \widetilde{\nabla}_\nu L_t, \nu_t \rangle.$$

In the following, we show an upper bound of the first part, and the proof for the second part follows similarly. Using the definition of the inner product,

$$\langle \nabla_\mu L(x_t) - \widetilde{\nabla}_\mu L_t, \mu_t \rangle = \sum_{i \in \mathcal{A}} \left( \nabla_{\mu,i} L(x_t) - \widetilde{\nabla}_{\mu,i} L_t \right) \cdot \mu_{t,i} \tag{12}$$

where $\mu_{t,i}$ is the $i$-th component in the vector $\mu_t$. Again, by taking the expectation of the biased gradient estimator,

$$\mathbb{E}[\widetilde{\nabla}_{\mu,i} L_t] = \mathbb{E}\left[ \frac{1 - R_t}{\mu_{t,i} + \gamma} \mathbb{I}\{a_t = i\} \right] = \sum_{j \in \mathcal{B}} \mathbb{P}(b_t = j) \cdot \mathbb{E}\left[ \frac{1 - R_t}{\mu_{t,i} + \gamma} \mathbb{I}\{a_t = i\} \right]$$
$$= \sum_{j \in \mathcal{B}} \nu_{t,j} \cdot \frac{l(i, j) \mu_{t,i}}{\mu_{t,i} + \gamma}.$$

Plugging it back to (12), and recalling the definitions that $\nabla_\mu L(x_t) = M\nu_t$ and $\nabla_{\mu,i} L(x_t) = \sum_{j\in\mathcal{B}} l(i,j)\nu_{t,j}$, we have that

$$\mathbb{E}\left[\langle \nabla_\mu L(x_t) - \widetilde{\nabla}_\mu L_t, \mu_t \rangle\right] = \sum_{i\in\mathcal{A}}\left(\sum_{j\in\mathcal{B}} l(i,j)\nu_{t,j} - \sum_{j\in\mathcal{B}} \nu_{t,j}\cdot\frac{l(i,j)\mu_{t,i}}{\mu_{t,i}+\gamma}\right)\cdot\mu_{t,i}$$

$$= \sum_{i\in\mathcal{A}}\sum_{j\in\mathcal{B}} l(i,j)\nu_{t,j}\left(1 - \frac{\mu_{t,i}}{\mu_{t,i}+\gamma}\right)\cdot\mu_{t,i}$$

$$= \sum_{i\in\mathcal{A}}\sum_{j\in\mathcal{B}} l(i,j)\nu_{t,j}\frac{\gamma}{1+\gamma/\mu_{t,i}}$$

$$\leq \sum_{i\in\mathcal{A}}\sum_{j\in\mathcal{B}} l(i,j)\nu_{t,j}\gamma$$

$$\leq A\gamma,$$

where the first inequality is due to $1+\gamma/\mu_{t,i} \geq 1$, and the second inequality uses the facts that $l(i,j) \leq 1, \forall i \in \mathcal{A}, j \in \mathcal{B}$ and $\sum_{j\in\mathcal{B}} \nu_{t,j} = 1$. A similar argument shows that $\mathbb{E}\left[\langle \nabla_\nu L(x_t) - \widetilde{\nabla}_\nu L_t, \nu_t \rangle\right] \leq B\gamma$. Combining the two parts, we conclude that the third term in (9) can be bounded by

$$-\mathbb{E}[\text{③}] = -\mathbb{E}[\langle\widetilde{\nabla} L_t - \nabla L(x_t), x_t\rangle] \leq (A+B)\gamma \leq 2A_{\max}\gamma. \tag{13}$$

From (9), we know that inequalities (10), (11), and (13) together can upper bound the expected value of the second term in (8). Finally, to bound the third term in (8), we apply Young's inequality and obtain almost surely that

$$\alpha_t\left\langle\widetilde{\nabla} L_t, x_t - x_{t+1}\right\rangle - D_\omega\left(x_{t+1}, x_t\right) = \alpha_t\left\langle\widetilde{\nabla} L_t, x_t - x_{t+1}\right\rangle - \frac{1}{2}\|x_{t+1} - x_t\|_2^2$$

$$\leq \frac{1}{2}\alpha_t^2\left\|\widetilde{\nabla} L_t\right\|_2^2 = \frac{1}{2}\alpha_t^2\left(\sum_{i\in\mathcal{A}}\frac{(1-R_t)^2\mathbb{I}\{a_t = i\}}{(\mu_{t,i}+\gamma)^2} + \sum_{j\in\mathcal{B}}\frac{(1-R_t)^2\mathbb{I}\{b_t = j\}}{(\nu_{t,j}+\gamma)^2}\right)$$

$$\leq \frac{A_{\max}\alpha_t^2}{\gamma^2}, \tag{14}$$

where the last step holds because $(\mu_{t,i}+\gamma)^2 \geq \gamma^2$, and $(1-R_t)^2\mathbb{I}\{a_t = i\} \leq 1$ almost surely. Substituting (10), (11), (13), and (14) back to (8), we finally conclude that

$$\mathbb{E}[L_{1/\lambda}(x_{t+1})] \leq \mathbb{E}[L_{1/\lambda}(x_t) + \lambda\alpha_t\left(L(\hat{x}_t) - L(x_t) + D_\omega(\hat{x}_t, x_t) + 2A_{\max}\gamma\right)] + \frac{A_{\max}\lambda\alpha_t^2}{\gamma^2}.$$

Rearranging, taking the sum from $t = 1$ to $T$, and telescoping leads to

$$\sum_{t=1}^{T}\mathbb{E}[\alpha_t(L(x_t) - L(\hat{x}_t) - D_\omega(\hat{x}_t, x_t))]$$

$$\leq \frac{1}{\lambda}\mathbb{E}[L_{1/\lambda}(x_1) - L_{1/\lambda}(x_{T+1})] + 2A_{\max}\gamma\sum_{t=1}^{T}\alpha_t + \frac{A_{\max}}{\gamma^2}\sum_{t=1}^{T}\alpha_t^2$$

$$\leq \frac{1}{\lambda} + 2A_{\max}\gamma\sum_{t=1}^{T}\alpha_t + \frac{A_{\max}}{\gamma^2}\sum_{t=1}^{T}\alpha_t^2, \tag{15}$$

where in the last step we used the fact that $0 \leq L_{1/\lambda}(x) \leq 1, \forall x \in \mathcal{X}$. In the following lemma, we show that bounding the LHS of (15) is sufficient to obtain an approximate stationary point.

**Lemma 7.** *Let $\hat{x} = prox_{1/\lambda, L}(x)$ with $\lambda > 1$. For any $\varepsilon > 0$, if $L(x) - L(\hat{x}) - D_\omega(\hat{x}, x) \leq \varepsilon$, then $x$ is a $\frac{2(2A_{\max}+\lambda)\sqrt{\varepsilon}}{\sqrt{\lambda-1}}$-approximate stationary point of the value function $V$.*

Dividing both sides of the inequality by $\sum_{t=1}^{T}\alpha_t$, we obtain that

$$\frac{\sum_{t=1}^{T}\mathbb{E}[\alpha_t(L(x_t) - L(\hat{x}_t) - D_\omega(\hat{x}_t, x_t))]}{\sum_{t=1}^{T}\alpha_t} \leq \frac{1 + \frac{A_{\max}\lambda}{\gamma^2}\sum_{t=1}^{T}\alpha_t^2}{\lambda\sum_{t=1}^{T}\alpha_t} + 2A_{\max}\gamma. \tag{16}$$

Recall that in Line 7 of Algorithm 1, the two agents uniformly sample a time index $\tau$ from $\{1, \ldots, T\}$ using a common random seed. Let $x_\tau = (\mu_\tau, \nu_\tau)$ be the output of the algorithm. Then, we can see that the LHS of (16) is equivalent to $\mathbb{E}[L(x_\tau) - L(\hat{x}_\tau) - D_\omega(\hat{x}_\tau, x_\tau)]$, because we assign equal values to the step sizes $\alpha_t$. Plugging back the parameter values $\alpha_t = T^{-3/4}$ and $\gamma = T^{-1/4}$, and letting $\lambda = 2$ yields,

$$\mathbb{E}[L(x_\tau) - L(\hat{x}_\tau) - D_\omega(\hat{x}_\tau, x_\tau)] \leq 4A_{\max}T^{-1/4}.$$

Together with Lemma 7, we can see that $x_\tau$ is a $16\sqrt{A_{\max}^3/T^{1/4}}$-approximate stationary point of $V$ in expectation. Finally, invoking Lemma 4, we can conclude that $x_\tau$ is an $16\sqrt{2A_{\max}^3/T^{1/4}}$-approximate Nash equilibrium in expectation. In other words, to obtain an $\varepsilon$-approximate NE, it suffices to use $O(A_{\max}^{12}/\varepsilon^8)$ samples. $\qquad\square$

### E.5  PROOF OF THEOREM 2

We first describe more explicitly the new variance-reduced algorithm that the agents are using, which has been skipped in the main text due to space limitations. Again, we only illustrate the algorithm for agent 1, as its counterpart for agent 2 is symmetric.

Similar to vanilla SGD (with no variance reduction) in Algorithm 1, agent 1 maintains a probability distribution $\mu_t$ over its action space for each time step $t \in [T]$. Such a distribution is uniformly initialized, i.e., $\mu_{1,i} = 1/A, \forall i \in \mathcal{A}$. At each time step $t \in [T]$, the agent first draws an action $a_t$ according to a perturbation of the distribution $\mu_t = (\mu_{t,1}, \ldots, \mu_{t,A})$ to be specified later. It then executes the selected action $a_t$ and observes the team reward $R_t$. Subsequently, the agent updates its probability distribution by following a variance-reduced gradient descent step.

The gradient update rule that we use and its convergence analysis are presented independently in Appendix F, as it can be considered as a solution to a standard constrained optimization problem on its own. Specifically, in Appendix F, we present a (projected) stochastic gradient descent algorithm that finds a stationary point in constrained smooth non-convex optimization at a rate of $O(1/T^{1/3})$. Our method is a straightforward extension of the STOchastic Recursive Momentum (STORM) algorithm proposed in Cutkosky & Orabona (2019), which uses a momentum-based approach to reduce the variances in SGD.

It is important to note that we rely on a non-adaptive variant of the STORM algorithm. In the original STORM (Algorithm 1 of Cutkosky & Orabona (2019)), the adaptive step size $\eta_t$ is calculated in a data-dependent manner (using knowledge of $\|\nabla f(x_t, \xi_t)\|$). This is ordinarily a desired property, as it alleviates the need of manual parameter tuning, and also finds wide applications in heuristic methods like Adam (Kingma & Ba, 2014). However, such adaptivity breaks our decentralized paradigm, because the agents update the step sizes using the local samples, and hence can end up with different step size values. This breaks the equivalence (between centralized and decentralized SGD) that we established in Lemma 5, which critically relies on equal step size values among all agents. Consequently, we need to refer to a non-adaptive variant of STORM (Cutkosky & Orabona, 2019) that replaces the data-dependent values with a universal upper bound $\sigma$.

Finally, to prove Theorem 2, we only need to instantiate the various notations ($F_t, \nabla f_t$, and $\xi_t$) of Appendix F in the context of the matrix team problem, and verify that all the assumptions in Appendix F are indeed satisfied. Then, we can complete the proof by invoking the convergence guarantee of Theorem 6 from Appendix F. It is easy to see that $F_t$ represents the loss function $L$ in Section 3, and the random variable $\xi_t$ is associated with the action drawn at time step $t$. The notation $\nabla f$ deserves more commentary. To ensure that the bounded variance assumption from Appendix F is satisfied, we cannot let the agent draw an action according to the distribution $\mu_t$, but instead according to a $\theta$-greedy parameterization of $\mu_t$ for some $\theta > 0$. Specifically, for a probability distribution $\mu_t$ at time $t$, the agent samples an action $a_t$ according to $\tilde{\mu}_t = (\tilde{\mu}_{t,1}, \ldots, \tilde{\mu}_{t,A})$, where:

$$\tilde{\mu}_{t,i} = (1 - \theta)\mu_{t,i} + \theta/A, \forall i \in \mathcal{A}.$$

To put it in a different way, with probability $1 - \theta$, the agent randomly selects an action according to the distribution $\mu_t$; with probability $\theta$, the agent uniformly samples $a_t$ from $\mathcal{A}$. The gradient estimator is instantiated with respect to the $\theta$-greedy parameterization as $\nabla f(\mu_t, \xi_t) = (\nabla_1 f(\mu_t, \xi_t), \ldots, \nabla_A f(\mu_t, \xi_t))$, where

$$\nabla_i f(\mu_t, \xi_t) = \frac{1 - R_t}{\tilde{\mu}_{t,i}}\mathbb{I}\{a_t = i\}, \forall i \in \mathcal{A}.$$

---

**Algorithm 4:** STORM (Cutkosky & Orabona, 2019) with Projections

---

1   $d_1 \leftarrow \nabla f(x_1, \xi_1)$;
2   **for** $t \leftarrow 1$ *to* $T$ **do**
3      $\eta_t \leftarrow \frac{k}{(w + \sigma^2 t)^{1/3}}$;
4      $x_{t+1} \leftarrow \Pi_{\mathcal{X}}(x_t - \eta_t d_t)$;
5      $a_{t+1} \leftarrow c\eta_t^2$;
6      $d_{t+1} \leftarrow \nabla f(x_{t+1}, \xi_{t+1}) + (1 - a_{t+1})(d_t - \nabla f(x_t, \xi_{t+1}))$;
7   Output $x_\tau$ where $\tau$ is uniformly sampled from $\{1, \ldots, T\}$;

---

We can see that $\nabla f(\mu_t, \xi_t)$ is an unbiased gradient estimator. Further, since $\tilde{\mu}_{t,i} \geq \theta/A, \forall i \in \mathcal{A}$, we can also verify that $\nabla f(\mu_t, \xi_t)$ has a bounded variance. Simple calculations show that Assumption 1 in Appendix F holds under the following instantiation of parameters:

$$\sigma^2 = \frac{2A_{\max}}{\theta}, \text{ and } L = \frac{4A_{\max}^2}{\theta^2}.$$

We can then invoke Theorem 6 to conclude that

$$\mathbb{E}\left[\left\|\frac{1}{\eta_\tau}(x_{\tau+1}^+ - x_\tau)\right\|^2\right] \leq \frac{Mw^{1/3}}{Tk} + \frac{M\sigma^{2/3}}{T^{2/3}k} \leq \frac{C\ln T}{T^{2/3}}\left(\frac{A_{\max}}{\theta}\right)^{11/3},$$

where $x_\tau = (\mu_\tau, \nu_\tau)$ and $C$ is some absolute constant. To ensure $\mathbb{E}\left[\left\|\frac{1}{\eta_\tau}(x_{\tau+1}^+ - x_\tau)\right\|\right] \leq \varepsilon$, it suffices that $T = \widetilde{\Omega}\left(\left(\frac{A}{\theta}\right)^{\frac{11}{2}} \cdot \frac{1}{\varepsilon^3}\right)$. If we further set $\theta = \varepsilon^{2/11}$, then we only need $T = \widetilde{\Omega}\left(\frac{A^{11/2}}{\varepsilon^4}\right)$. From the gradient mapping property (Lemma 3), we know that $\left\|\frac{1}{\eta_\tau}(x_{\tau+1}^+ - x_\tau)\right\| \leq \varepsilon$ implies a $\frac{5}{4}\varepsilon$-approximate stationary point, where we used the condition that $\eta_\tau \leq \frac{1}{4L}$ in Appendix F. Finally, we invoke Lemma 4 to conclude with an $\varepsilon$-approximate Nash equilibrium after $T = \widetilde{\Omega}(1/\varepsilon^4) \cdot \text{poly}(A_{\max})$ rounds. This completes the proof of Theorem 2.

## F   SGD WITH VARIANCE REDUCTION

In this section, we present a (projected) stochastic gradient descent method with variance reduction (Johnson & Zhang, 2013; Allen-Zhu & Hazan, 2016; Reddi et al., 2016). This method proceeds by adding a simple projection step to a non-adaptive variant of the STOchastic Recursive Momentum (STORM) algorithm proposed in Cutkosky & Orabona (2019).

We consider a generic stochastic non-convex optimization problem as follows: We are given an objective function $F : \mathbb{R}^A \to \mathbb{R}$, and our goal is to find a point $x \in \mathcal{X} \subseteq \mathbb{R}^A$ such that $\nabla F(x)$ is close to 0, where $\mathcal{X}$ is the feasible region. We do not have accurate information about the function $F$, and can only access it through a stochastic sampling oracle $f(\cdot, \xi)$, where the random variable $\xi$ represents the "randomness" of the oracle. Throughout this section, we make the following assumptions that are standard in the literature (Arjevani et al., 2019). All norms $\|\cdot\|$ in this section are $L^2$ norms unless otherwise specified.

**Assumption 1.** *1. We have access to a stream of random variables $\xi_1, \ldots, \xi_T$, such that the gradient estimators are unbiased and have bounded variances: $\nabla \mathbb{E}_{\xi_t}[f(x, \xi_t)] = \nabla F(x)$, and $\mathbb{E}[\|\nabla f(x, \xi_t) - \nabla F(x)\|^2] \leq \sigma^2$ for some $\sigma > 0$ for all $t \in [T]$ and $x \in \mathcal{X}$.*

*2. The objective $F$ has bounded initial sub-optimality and is $L$-smooth: $F(x_0) - \inf_{x \in \mathcal{X}} F(x) < \infty$, and $\|\nabla F(x) - \nabla F(y)\| \leq L \cdot \|x - y\|, \forall x, y \in \mathbb{R}^A$ for some $L > 0$. The stochastic oracle also satisfies a mean-squared smoothness property for the same constant $L$: $\mathbb{E}[\|\nabla f(x, \xi) - \nabla f(y, \xi)\|^2] \leq L^2 \cdot \|x - y\|^2, \forall x, y \in \mathbb{R}^A$ with probability 1.*

Our method STORM with projections is formally presented in Algorithm 4. STORM uses a variant of momentum to reduce the variance of the gradients in SGD. It achieves an optimal convergence rate of $O(1/T^{1/3})$, which improves over the standard convergence rate $O(1/T^{1/4})$ of SGD with no

variance reduction. Before we present the convergence guarantee of Algorithm 4, we first introduce a few notations for ease of presentations.

For any $t \in [T]$, we break the update rule into two steps:

$$\tilde{x}_{t+1} \stackrel{\text{def}}{=} x_t - \eta_t d_t, \text{ and } x_{t+1} = \Pi_{\mathcal{X}}(\tilde{x}_{t+1}).$$

In addition, for each $t \in [T]$, we define $x_{t+1}^+ \stackrel{\text{def}}{=} \Pi_{\mathcal{X}}(x_t - \eta_t \nabla F(x_t))$ to be the next iterate updated using the full gradient $\nabla F(x_t)$, a value we do not have access to. Define $\varepsilon_t \stackrel{\text{def}}{=} d_t - \nabla F(x_t)$ to be the error in $d_t$. The high-level procedure of our proof is to seek to upper bound the value $\mathbb{E}\left[\sum_{t=1}^T \left\| \frac{1}{\eta_t} \left( x_{t+1}^+ - x_t \right) \right\|^2\right]$, and then to invoke the gradient mapping property in Lemma 3 to conclude with a stationary point. This is in contrast with the unconstrained case, where Cutkosky & Orabona (2019) directly derive an upper bound of $\mathbb{E}\left[\sum_{t=1}^T \|\nabla F(x_t)\|^2\right]$. In the following, we start with a few technical lemmas.

**Lemma 8.** *Suppose $\eta_t \leq \frac{1}{4L}$ for all $t \in [T]$. Then,*

$$\mathbb{E}[F(x_{t+1}) - F(x_t)] \leq \mathbb{E}\left[-\frac{3}{16\eta_t} \left\| x_{t+1}^+ - x_t \right\|^2 + \frac{7\eta_t}{8} \|\varepsilon_t\|^2\right].$$

*Proof.* From the first-order optimality condition, we know that

$$\langle x - x_{t+1}, x_{t+1} - (x_t - \eta_t d_t) \rangle \geq 0,$$

for any $x \in \mathcal{X}$. Taking $x = x_t$ leads to

$$\langle x_t - x_{t+1}, x_{t+1} - x_t \rangle + \langle x_t - x_{t+1}, \eta_t d_t \rangle \geq 0,$$

which in turn implies that

$$\langle x_{t+1} - x_t, d_t \rangle \leq -\frac{1}{\eta_t} \|x_t - x_{t+1}\|^2. \tag{17}$$

It follows that

$$\begin{aligned}
\langle \nabla F(x_t), x_{t+1} - x_t \rangle &= \langle d_t - \varepsilon_t, x_{t+1} - x_t \rangle \\
&\leq -\frac{1}{\eta_t} \|x_t - x_{t+1}\|^2 - \langle \varepsilon_t, x_{t+1} - x_t \rangle \\
&\leq -\frac{1}{\eta_t} \|x_t - x_{t+1}\|^2 + \frac{\eta_t}{2} \|\varepsilon_t\|^2 + \frac{1}{2\eta_t} \|x_{t+1} - x_t\|^2 \\
&= -\frac{1}{2\eta_t} \|x_t - x_{t+1}\|^2 + \frac{\eta_t}{2} \|\varepsilon_t\|^2,
\end{aligned}$$

where the first inequality uses (17), and the second inequality is due to Hölder's inequality and Young's inequality. From the smoothness of $F$,

$$\begin{aligned}
\mathbb{E}[F(x_{t+1})] &\leq \mathbb{E}\left[F(x_t) + \langle \nabla F(x_t), x_{t+1} - x_t \rangle + \frac{L}{2} \|x_{t+1} - x_t\|^2\right] \\
&\leq \mathbb{E}\left[F(x_t) - \frac{1}{2\eta_t} \|x_t - x_{t+1}\|^2 + \frac{\eta_t}{2} \|\varepsilon_t\|^2 + \frac{L}{2} \|x_{t+1} - x_t\|^2\right] \\
&\leq \mathbb{E}\left[F(x_t) - \frac{3}{8\eta_t} \|x_t - x_{t+1}\|^2 + \frac{\eta_t}{2} \|\varepsilon_t\|^2\right], \tag{18}
\end{aligned}$$

where the last step uses $\eta_t \leq \frac{1}{4L}$. From the fact that $\|x + y\|^2 \leq 2\|x\|^2 + 2\|y\|^2$, we know

$$\left\| x_{t+1}^+ - x_t \right\|^2 = \left\| x_{t+1}^+ - x_{t+1} + x_{t+1} - x_t \right\|^2 \leq 2 \left\| x_{t+1}^+ - x_{t+1} \right\|^2 + 2 \|x_{t+1} - x_t\|^2.$$

Rearranging the terms leads to

$$
-\|x_t - x_{t+1}\|^2 \le \|x_{t+1}^+ - x_{t+1}\|^2 - \frac{1}{2}\|x_{t+1}^+ - x_t\|^2
$$

$$
\le \|\Pi_{\mathcal{X}}(x_t - \eta_t \nabla F(x_t)) - \Pi_{\mathcal{X}}(x_t - \eta_t d_t)\|^2 - \frac{1}{2}\|x_{t+1}^+ - x_t\|^2
$$

$$
\le \|(x_t - \eta_t \nabla F(x_t)) - (x_t - \eta_t d_t)\|^2 - \frac{1}{2}\|x_{t+1}^+ - x_t\|^2
$$

$$
= \eta_t^2 \|\varepsilon_t\|^2 - \frac{1}{2}\|x_{t+1}^+ - x_t\|^2 . \tag{19}
$$

The second inequality uses the definition of $x_{t+1}^+$. The third step holds because the projection operator is non-expansive, i.e. $\|\Pi_{\mathcal{X}}(x) - \Pi_{\mathcal{X}}(y)\| \le \|x - y\|$. Substituting (19) back to (18) leads to

$$
\mathbb{E}[F(x_{t+1})] \le \mathbb{E}\left[F(x_t) - \frac{3}{16\eta_t}\|x_{t+1}^+ - x_t\|^2 + \frac{7\eta_t}{8}\|\varepsilon_t\|^2\right].
$$

Rearranging the terms completes the proof. $\qquad\square$

**Lemma 9.** *(Lemma 3 in Cutkosky & Orabona (2019)). For any $t \in [T]$, it holds that*

$$
\mathbb{E}\left[\frac{(1-a_t)^2}{\eta_{t-1}}(\nabla f(x_t, \xi_t) - \nabla F(x_t)) \cdot \varepsilon_{t-1}\right] = 0,
$$

$$
\mathbb{E}\left[\frac{(1-a_t)^2}{\eta_{t-1}}(\nabla f(x_t, \xi_t) - \nabla f(x_{t-1}, \xi_t) - \nabla F(x_t) + \nabla F(x_{t-1})) \cdot \varepsilon_{t-1}\right] = 0.
$$

**Lemma 10.** *(Adapted from Lemma 5 in Cutkosky & Orabona (2019)). With the notations in Algorithm 4, we have*

$$
\mathbb{E}\left[\frac{\|\varepsilon_t\|^2}{\eta_{t-1}}\right] \le \mathbb{E}\left[2c^2\eta_{t-1}^3\sigma^2 + \frac{(4L^2\eta_{t-1}^2 + 1)(1-a_t)^2}{\eta_{t-1}}\|\varepsilon_{t-1}\|^2 + \frac{4(1-a_t)^2 L^2}{\eta_{t-1}}\|x_t^+ - x_{t-1}\|^2\right].
$$

*Proof.* First, observe that

$$
\mathbb{E}\left[\|\nabla f(x_t, \xi_t) - \nabla f(x_{t-1}, \xi_t) - \nabla F(x_t) + \nabla F(x_{t-1})\|^2\right]
$$

$$
\le \mathbb{E}\left[\|\nabla f(x_t, \xi_t) - \nabla f(x_{t-1}, \xi_t)\|^2 + \|\nabla F(x_t) - \nabla F(x_{t-1})\|^2\right]
$$

$$
- 2\mathbb{E}\left[\langle \nabla f(x_t, \xi_t) - \nabla f(x_{t-1}, \xi_t), \nabla F(x_t) - \nabla F(x_{t-1})\rangle\right]
$$

$$
\le \mathbb{E}\left[\|\nabla f(x_t, \xi_t) - \nabla f(x_{t-1}, \xi_t)\|^2 + \|\nabla F(x_t) - \nabla F(x_{t-1})\|^2\right]
$$

$$
- 2\mathbb{E}\left[\mathbb{E}\left[\langle \nabla f(x_t, \xi_t) - \nabla f(x_{t-1}, \xi_t), \nabla F(x_t) - \nabla F(x_{t-1})\rangle \mid \xi_1, \dots, \xi_{t-1}\right]\right]
$$

$$
= \mathbb{E}\left[\|\nabla f(x_t, \xi_t) - \nabla f(x_{t-1}, \xi_t)\|^2 - \|\nabla F(x_t) - \nabla F(x_{t-1})\|^2\right]
$$

$$
\le \mathbb{E}\left[\|\nabla f(x_t, \xi_t) - \nabla f(x_{t-1}, \xi_t)\|^2\right]. \tag{20}
$$

By the definition of $\varepsilon_t$, we have $\varepsilon_t = d_t - \nabla F(x_t) = \nabla f(x_t, \xi_t) + (1-a_t)(d_{t-1} - \nabla f(x_{t-1}, \xi_t)) - \nabla F(x_t)$. Therefore,

$$
\mathbb{E}\left[\frac{\|\varepsilon_t\|^2}{\eta_{t-1}}\right] = \mathbb{E}\left[\frac{1}{\eta_{t-1}}\|\nabla f(x_t, \xi_t) + (1-a_t)(d_{t-1} - \nabla f(x_{t-1}, \xi_t)) - \nabla F(x_t)\|^2\right]
$$

$$
= \mathbb{E}\left[\frac{1}{\eta_{t-1}}\|a_t(\nabla f(x_t, \xi_t) - \nabla F(x_t)) + (1-a_t)(d_{t-1} - \nabla F(x_{t-1}))\right.
$$

$$
\left. + (1-a_t)(\nabla f(x_t, \xi_t) - \nabla f(x_{t-1}, \xi_t) - \nabla F(x_t) + \nabla F(x_{t-1}))\|^2\right]
$$

$$
\le \mathbb{E}\left[2c^2\eta_{t-1}^3\|\nabla f(x_t, \xi_t) - \nabla F(x_t)\|^2 + \frac{1}{\eta_{t-1}}(1-a_t)^2\|\varepsilon_{t-1}\|^2\right.
$$

$$
\left. + \frac{2}{\eta_{t-1}}(1-a_t)^2\|\nabla f(x_t, \xi_t) - \nabla f(x_{t-1}, \xi_t) - \nabla F(x_t) + \nabla F(x_{t-1})\|^2\right],
$$

where in the last step we used Lemma 9 and the simple fact that $\|x + y\|^2 \leq 2 \|x\|^2 + 2 \|y\|^2$. Further applying (20) and the assumption that $\mathbb{E}\left[\|\nabla f(x_t, \xi_t) - \nabla F(x_t)\|^2\right] \leq \sigma^2$ leads to

$$
\begin{aligned}
\mathbb{E}\left[\frac{\|\varepsilon_t\|^2}{\eta_{t-1}}\right] =& \mathbb{E}\left[2c^2\eta_{t-1}^3\sigma^2 + \frac{(1-a_t)^2}{\eta_{t-1}}\|\varepsilon_{t-1}\|^2 + \frac{2(1-a_t)^2}{\eta_{t-1}}\|\nabla f(x_t,\xi_t) - \nabla f(x_{t-1},\xi_t)\|^2\right] \\
\leq& \mathbb{E}\left[2c^2\eta_{t-1}^3\sigma^2 + \frac{(1-a_t)^2}{\eta_{t-1}}\|\varepsilon_{t-1}\|^2 + \frac{2(1-a_t)^2 L^2}{\eta_{t-1}}\|x_t - x_{t-1}\|^2\right] \\
\leq& \mathbb{E}\left[2c^2\eta_{t-1}^3\sigma^2 + \frac{(1-a_t)^2}{\eta_{t-1}}\|\varepsilon_{t-1}\|^2 + \frac{4(1-a_t)^2 L^2}{\eta_{t-1}}\left(\|x_t - x_t^+\|^2 + \|x_t^+ - x_{t-1}\|^2\right)\right] \\
\leq& \mathbb{E}\left[2c^2\eta_{t-1}^3\sigma^2 + \frac{(4L^2\eta_{t-1}^2 + 1)(1-a_t)^2}{\eta_{t-1}}\|\varepsilon_{t-1}\|^2 + \frac{4(1-a_t)^2 L^2}{\eta_{t-1}}\|x_t^+ - x_{t-1}\|^2\right].
\end{aligned}
$$

The first inequality is due to the $L$-smoothness of the function $f$. The second inequality again uses the fact that $\|x + y\|^2 \leq 2 \|x\|^2 + 2 \|y\|^2$. The last step holds because of the non-expansiveness of the projection operator, that is,

$$
\|x_t - x_t^+\| = \|\Pi_{\mathcal{X}}(x_{t-1} - \eta_{t-1}d_{t-1}) - \Pi_{\mathcal{X}}(x_{t-1} - \eta_{t-1}\nabla F(x_{t-1}))\| \leq \eta_{t-1}^2 \|\varepsilon_{t-1}\|^2.
$$

This completes the proof of the lemma. $\qquad\square$

Now, we are ready to present the convergence guarantee of Algorithm 4.

**Theorem 6.** *(Adapted from Theorem 2 in Cutkosky & Orabona (2019)). Suppose the conditions in Assumption 1 are satisfied. For any $b > 0$, let $k = \frac{b\sigma^{\frac{2}{3}}}{L}, c = 32L^2 + \sigma^2/\left(7Lk^3\right) = L^2\left(32 + 1/\left(7b^3\right)\right), w = \max\left((4Lk)^3, 2\sigma^2, \left(\frac{ck}{4L}\right)^3\right) = \sigma^2\max\left((4b)^3, 2, \left(32b + \frac{1}{7b^2}\right)^3/64\right),$ and $M = 16(F(x_1) - F^\star) + \frac{w^{1/3}\sigma^2}{2L^2 k} + \frac{k^3 c^2}{L^2}\ln(T + 2)$. Then, the following convergence guarantee holds for Algorithm 4*

$$
\mathbb{E}\left[\frac{1}{T}\sum_{t=1}^{T}\left\|\frac{1}{\eta_t}(x_{t+1}^+ - x_t)\right\|^2\right] \leq \frac{Mw^{1/3}}{Tk} + \frac{M\sigma^{2/3}}{T^{2/3}k}.
$$

*Proof.* The proof is similar to that of Theorem 2 in Cutkosky & Orabona (2019), and we reproduce it here for convenience of the reader. First, define the Lyapunov function $\Phi_t = F(x_t) + \frac{1}{32L^2\eta_{t-1}}\|\varepsilon_t\|^2$. From Lemma 10, we can derive that

$$
\begin{aligned}
&\mathbb{E}\left[\frac{\|\varepsilon_{t+1}\|^2}{\eta_t} - \frac{\|\varepsilon_t\|^2}{\eta_{t-1}}\right] \\
\leq& \mathbb{E}\left[2c^2\eta_t^3\sigma^2 + \frac{(4L^2\eta_t^2 + 1)(1 - a_{t+1})^2}{\eta_t}\|\varepsilon_t\|^2 + \frac{4(1 - a_{t+1})^2 L^2}{\eta_t}\|x_{t+1}^+ - x_t\|^2 - \frac{\|\varepsilon_t\|^2}{\eta_{t-1}}\right] \\
\leq& \mathbb{E}\left[\underbrace{2c^2\eta_t^3\sigma^2}_{A_t} + \underbrace{\left(\frac{(4L^2\eta_t^2 + 1)(1 - a_{t+1})^2}{\eta_t} - \frac{1}{\eta_{t-1}}\right)\|\varepsilon_t\|^2}_{B_t} + \underbrace{\frac{4(1 - a_{t+1})^2 L^2}{\eta_t}\|x_{t+1}^+ - x_t\|^2}_{C_t}\right].
\end{aligned}
$$

The first two terms $A_t$ and $B_t$ are exactly the same as in the proof of Theorem 2 in Cutkosky & Orabona (2019), and we refer to their results as follows:

$$
\sum_{t=1}^{T} A_t \leq 2k^3 c^2 \ln(T + 2), \text{ and } \sum_{t=1}^{T} B_t \leq -28L^2 \sum_{t=1}^{T} \eta_t \|\varepsilon_t\|^2.
$$

From $w \geq (4Lk)^3$, we know that $\eta_t \leq \frac{1}{4L}$. Further, since $a_{t+1} = c\eta_t^2$, we have that $a_{t+1} \leq \frac{ck}{4Lw^{1/3}} \leq 1$ for all $t$, and hence $C_t \leq \frac{4L^2}{\eta_t} \left\| x_{t+1}^+ - x_t \right\|^2$. Putting it all together, we obtain

$$\frac{1}{32L^2} \sum_{t=1}^{T} \left( \frac{\|\varepsilon_{t+1}\|^2}{\eta_t} - \frac{\|\varepsilon_t\|^2}{\eta_{t-1}} \right) \leq \frac{k^3 c^2}{16L^2} \ln(T+2) + \sum_{t=1}^{T} \left( \frac{1}{8\eta_t} \left\| x_{t+1}^+ - x_t \right\|^2 - \frac{7\eta_t}{8} \|\varepsilon_t\|^2 \right).$$

(21)

From Lemma 8, we know that

$$\mathbb{E}\left[\Phi_{t+1} - \Phi_t\right] \leq \mathbb{E}\left[ -\frac{3}{16\eta_t} \left\| x_{t+1}^+ - x_t \right\|^2 + \frac{7\eta_t}{8} \|\varepsilon_t\|^2 + \frac{1}{32L^2\eta_t} \|\varepsilon_{t+1}\|^2 - \frac{1}{32L^2\eta_{t-1}} \|\varepsilon_t\|^2 \right].$$

Summing over $t$ from 1 to $T$ and then applying (21), we obtain

$$\mathbb{E}\left[\Phi_{T+1} - \Phi_1\right] \leq \sum_{t=1}^{T} \mathbb{E}\left[ -\frac{3}{16\eta_t} \left\| x_{t+1}^+ - x_t \right\|^2 + \frac{7\eta_t}{8} \|\varepsilon_t\|^2 + \frac{1}{32L^2\eta_t} \|\varepsilon_{t+1}\|^2 - \frac{1}{32L^2\eta_{t-1}} \|\varepsilon_t\|^2 \right]$$

$$\leq \mathbb{E}\left[ \frac{k^3 c^2}{16L^2} \ln(T+2) - \sum_{t=1}^{T} \frac{1}{16\eta_t} \left\| x_{t+1}^+ - x_t \right\|^2 \right].$$

Rearranging the terms leads to

$$\mathbb{E}\left[ \sum_{t=1}^{T} \frac{1}{\eta_t} \left\| x_{t+1}^+ - x_t \right\|^2 \right] \leq \mathbb{E}\left[ 16(\Phi_1 - \Phi_{T+1}) + \frac{k^3 c^2}{L^2} \ln(T+2) \right]$$

$$\leq 16(F(x_1) - F^\star) + \frac{1}{2L^2\eta_0} \mathbb{E}[\|\varepsilon_1\|^2] + \frac{k^3 c^2}{L^2} \ln(T+2)$$

$$\leq 16(F(x_1) - F^\star) + \frac{w^{1/3}\sigma^2}{2L^2 k} + \frac{k^3 c^2}{L^2} \ln(T+2),$$

where the last step holds due to the definition that $\eta_0 = \frac{k}{w^{1/3}}$. Since $\eta_t$ is decreasing in $t$,

$$\mathbb{E}\left[ \sum_{t=1}^{T} \frac{1}{\eta_t} \left\| x_{t+1}^+ - x_t \right\|^2 \right] = \mathbb{E}\left[ \sum_{t=1}^{T} \eta_t \left\| \frac{1}{\eta_t} (x_{t+1}^+ - x_t) \right\|^2 \right] \geq \eta_T \mathbb{E}\left[ \sum_{t=1}^{T} \left\| \frac{1}{\eta_t} (x_{t+1}^+ - x_t) \right\|^2 \right].$$

Dividing both sides by $T\eta_T$ and recalling the definition $M = 16(F(x_1) - F^\star) + \frac{w^{1/3}\sigma^2}{2L^2 k} + \frac{k^3 c^2}{L^2} \ln(T+2)$, we obtain

$$\mathbb{E}\left[ \frac{1}{T} \sum_{t=1}^{T} \left\| \frac{1}{\eta_t} (x_{t+1}^+ - x_t) \right\|^2 \right] \leq \frac{M}{T\eta_T} = \frac{M(w + \sigma^2 T)^3}{Tk} \leq \frac{Mw^{1/3}}{Tk} + \frac{M\sigma^{2/3}}{T^{2/3} k},$$

where in the last step we used the fact that $(a + b)^{1/3} \leq a^{1/3} + b^{1/3}$. $\square$

## G  MISSING RESULTS IN SECTION 5

### G.1  PROOF OF LEMMA 1

In the following, we give a proof for the first inequality. The second inequality can be proved via a similar argument. The desired result clearly holds for any state $s$ that is in its first stage, due to our definition of $V_h^{\star, \bar{\nu}_h^k}(s)$ for this special case. In the following, we only need to focus on the case where $\overline{V}_h(s)$ has been updated at least once at the given state $s$ before the $k$-th episode.

Our proof relies on induction on $k \in [K]$. First, the claim holds for $k = 1$ due to the aforementioned logic. For each step $h \in [H]$ and $s \in \mathcal{S}$, we consider the following two cases.

**Case 1:** $\overline{V}_h(s)$ has just been updated in (the end of) episode $k - 1$. In this case,

$$\overline{V}_h^k(s) = \frac{1}{\check{n}} \sum_{i=1}^{\check{n}} \left( r_h(s, a_h^{\check{l}_i}, b_h^{\check{l}_i}) + \overline{V}_{h+1}^{\check{l}_i}(s_{h+1}^{\check{l}_i}) \right) + b_{\check{n}}.$$

(22)

By the definition of $V_h^{\star,\bar{\nu}_h^k}(s)$ and the induction hypothesis,

$$
\begin{aligned}
V_h^{\star,\bar{\nu}_h^k}(s) =& \frac{1}{\check{n}} \sum_{i=1}^{\check{n}} \max_{\mu} \mathbb{D}_{\mu \times \nu_h^{\check{l}_i}} \left( r_h + \mathbb{P}_h V_{h+1}^{\star,\bar{\nu}_{h+1}^{\check{l}_i}} \right)(s) \\
\leq& \frac{1}{\check{n}} \sum_{i=1}^{\check{n}} \max_{\mu} \mathbb{D}_{\mu \times \nu_h^{\check{l}_i}} \left( r_h + \mathbb{P}_h \overline{V}_{h+1}^{\check{l}_i} \right)(s),
\end{aligned}
\tag{23}
$$

Further, we define the regret

$$
R_{\check{n}}(s) = \frac{1}{\check{n}} \sum_{i=1}^{\check{n}} \max_{\mu} \mathbb{D}_{\mu \times \nu_h^{\check{l}_i}} \left( r_h + \mathbb{P}_h \overline{V}_{h+1}^{\check{l}_i} \right)(s) - \frac{1}{\check{n}} \sum_{i=1}^{\check{n}} \mathbb{D}_{\mu_h^{\check{l}_i} \times \nu_h^{\check{l}_i}} \left( r_h + \mathbb{P}_h \overline{V}_{h+1}^{\check{l}_i} \right)(s). \tag{24}
$$

Since we only update the optimistic value function $\overline{V}_h$ at the end of a stage, its value remain unchanged within each stage. Observe that in (24), the iterator $\check{l}_i$ takes values of episode indices that belong to the same stage. Therefore, we know that $\overline{V}_{h+1}^{\check{l}_i}$ is a constant value for all $i \in \{1, \ldots, \check{n}\}$. We can hence drop the index $i$ and simply rewrite it as $\overline{V}_{h+1}^{\check{l}}$ when there is no ambiguity. With such formulation, one may quickly realize that bounding the regret $R_{\check{n}}(s)$ reduces to analyzing a matrix team problem at each state $s$, as we have defined in Section 3. Specifically, the matrix team lasts for $\check{n}$ rounds. For a strategy pair $(\mu, \nu)$ of the players, the expected reward function for this matrix team is $\mathbb{D}_{\mu \times \nu}(r_h + \mathbb{P}_h \overline{V}_{h+1}^{\check{l}})(s)$.

**Remark 2.** *It is worth mentioning that an additional step is required if one uses V-learning with the celebrated learning rate $\alpha_t = \frac{H+1}{H+t}$ (Jin et al., 2018) instead to update $\overline{V}_h$, which induces an update rule as follows:*

$$
\overline{V}_h(s_h) \leftarrow (1 - \alpha_t) \overline{V}_h(s_h) + \alpha_t \left( r_h(s_h, a_h, b_h) + \overline{V}_{h+1}(s_{h+1}) + \beta_t \right), \tag{25}
$$

*where $t$ is the number of times that $s_h$ has been visited, and $\beta_t$ is some bonus term. In this way, $\overline{V}_h$ is updated every time the state $s_h$ is visited, and hence its value varies at each step. Drawing connections back to our approach, this will lead to a matrix team problem where the expected reward function is non-stationary over time. Our algorithm and analyses in Section 3 do not directly apply to such a non-stationary problem. However, in our experience, we were able to show that it is possible to carefully tune the order of the learning rate $\alpha$ in a way such that the level of non-stationarity in $\overline{V}_h$ can be "endured" (guarantees sublinear regret in (24)) by a specifically-designed non-stationary variant of Algorithm 1. Therefore, in this step, the learning rate $\alpha_t = \frac{H+1}{H+t}$ requires more delicate treatments but does not completely fail.*

Since in Algorithm 2, both players essentially run an individual copy of Algorithm 1 for each state, our analysis exactly reduces to the investigation of Algorithm 1 in the matrix team problem. One may also realize that taking the average over $\check{n}$ in (24) is equivalent to uniformly sample a time index $\tau$ from $\{1, \ldots, \check{n}\}$ and then take the expectation over the randomness of such a sampling process. This is exactly how we constructed the approximate Nash equilibrium in Theorem 1. By applying the results (and proofs) of Theorem 1, we deduce that $(\mu_h^{\check{l}_\tau}, \nu_h^{\check{l}_\tau})$ is an $16H\sqrt{2A_{\max}^3/\check{n}^{1/4}}$-approximate NE in expectation after running Algorithm 1 for $\check{n}$ rounds. The extra $H$ factor comes from the additional nuance that in the problem formulation of Theorem 1, the reward is assumed to be bounded in $[0, 1]$, while in the current matrix team problem, the rewards lie in $[0, H]$. The result above implies that for any $\mu_\tau^\dagger \in \Delta(\mathcal{A})$ that is a best response of $\nu_h^{\check{l}_\tau}$, it holds that

$$
\mathbb{E} \left[ \mathbb{D}_{\mu_\tau^\dagger \times \nu_h^{\check{l}_\tau}}(r_h + \mathbb{P}_h \overline{V}_{h+1}^{\check{l}})(s) - \mathbb{D}_{\mu_h^{\check{l}_\tau} \times \nu_h^{\check{l}_\tau}}(r_h + \mathbb{P}_h \overline{V}_{h+1}^{\check{l}})(s) \right] \leq 16H\sqrt{2A_{\max}^3/\check{n}^{1/4}},
$$

where the expectation is taken over the randomness of $\tau$, the rewards, and the algorithm itself. The above inequality is equivalent to

$$
\mathbb{E}[R_{\check{n}}] \leq 16H\sqrt{2A_{\max}^3/\check{n}^{1/4}}, \tag{26}
$$

where the expectation is only with respect to the randomness of the rewards and the algorithm.

**Remark 3.** *Again, we would like to discuss the alternative of using V-learning with the popular learning rate $\alpha_t = \frac{H+1}{H+t}$ (Jin et al., 2018). With such a learning rate, the update rule (25) of $\overline{V}_h$ can be equivalently expressed as*

$$\overline{V}_h^k(s) = \alpha_t^0 H + \sum_{i=1}^t \alpha_t^i \left[ r_h\left(s, a_h^{k^i}, b_h^{k^i}\right) + \overline{V}_{h+1}^{k^i}\left(s_{h+1}^{k^i}\right) + \beta_i \right],$$

*where $t$ is the number of times that $s_h$ has been visited, $k^i$ is the index of the episode such that $s_h$ is visited the $i$-th time, and $\beta_i$ is some bonus term. The weights $\alpha_t^i$ are given by*

$$\alpha_t^0 = \prod_{j=1}^t \left(1 - \alpha_j\right), \quad and \quad \alpha_t^i = \alpha_i \prod_{j=i+1}^t \left(1 - \alpha_j\right), \forall 1 \le i \le t.$$

*If we redefine $V_h^{\bar{\mu}_h^k, \star}(s)$ and $V_h^{\star, \bar{\nu}_h^k}(s)$ to comply with this update rule, the regret in (24) would be rewritten as*

$$R_t(s) = \sum_{i=1}^t \alpha_t^i \max_\mu \mathbb{D}_{\mu \times \nu_h^{ki}}\left(r_h + \mathbb{P}_h \overline{V}_{h+1}^{k^i}\right)(s) - \sum_{i=1}^t \alpha_t^i \mathbb{D}_{\mu_h^{ki} \times \nu_h^{ki}}\left(r_h + \mathbb{P}_h \overline{V}_{h+1}^{k^i}\right)(s).$$

*For now, let us forget about the non-stationarity of $\overline{V}_{h+1}^{k^i}$, and pretend that it can be considered as a constant term throughout the entire horizon by applying the method sketched in Remark 2. Even in this simplified scenario, for each fixed $t$, we still need to address a weighted matrix team problem where the reward at time $i$ is assigned the weight $\alpha_t^i$. Further, as $t$ varies, the weight $\alpha_t^i$ assigned to the same step $i$ also changes over time. Unfortunately, such dynamically weighted rewards cannot be easily handled by Algorithm 1, as well as other stochastic non-convex optimization algorithms that we are aware of. During the execution of Algorithm 1, we are not able to modify the learning step size of a previous step online to cope with the changing weights. These weights also cannot be pre-computed, because it relies on knowing the total number of times that a certain state $s_h$ is visited during the entire horizon, which is impossible before seeing the output of the algorithm. Therefore, such an SGD framework is incompatible with the dynamic weights $\alpha_t^i$ that change over $t$, and this technical difficulty prevents us from directly applying the popular learning rate $\alpha_t = \frac{H+1}{H+t}$. Instead, we proposed to utilize stage-based V-learning, because it essentially assigns uniform weights to each time step in history. These weights only depend on the number of visitations to the state $s_h$ in a certain stage, which can be pre-computed from the recursive formula $e_{i+1} = \lfloor (1 + \frac{1}{H})e_i \rfloor$.*

Further, let $\mathcal{F}_i$ be the $\sigma$-algebra generated by all the random variables before episode $\breve{l}_i$. Then, we can see that $\{r_h(s, a_h^{\breve{l}_i}, b_h^{\breve{l}_i}) + \overline{V}_{h+1}^{\breve{l}_i}(s_{h+1}^{\breve{l}_i})\}_{i=1}^{\breve{n}}$ is a martingale with respect to $\{\mathcal{F}_i\}_{i=1}^{\breve{n}}$. For any probability $p \in (0, 1]$, let $\iota = \log(2SKH/p)$. From the Azuma-Hoeffding inequality, it holds with probability at least $1 - p/(2SHK)$ that

$$\frac{1}{\breve{n}} \sum_{i=1}^{\breve{n}} \mathbb{D}_{\mu_h^{\breve{l}_i} \times \nu_h^{\breve{l}_i}}\left(r_h + \mathbb{P}_h \overline{V}_{h+1}^{\breve{l}_i}\right)(s) - \frac{1}{\breve{n}} \sum_{i=1}^{\breve{n}} \left(r_h(s, a_h^{\breve{l}_i}, b_h^{\breve{l}_i}) + \overline{V}_{h+1}^{\breve{l}_i}(s_{h+1}^{\breve{l}_i})\right) \le 2H\sqrt{\iota/\breve{n}}, \quad (27)$$

where $\iota$ suppresses logarithmic terms. In fact, since we only need Lemma 1 to hold in expectation instead of with high probability, it suffices to replace (27) with an even simpler statement:

$$\mathbb{E}\left[ \frac{1}{\breve{n}} \sum_{i=1}^{\breve{n}} \mathbb{D}_{\mu_h^{\breve{l}_i} \times \nu_h^{\breve{l}_i}}\left(r_h + \mathbb{P}_h \overline{V}_{h+1}^{\breve{l}_i}\right)(s) - \frac{1}{\breve{n}} \sum_{i=1}^{\breve{n}} \left(r_h(s, a_h^{\breve{l}_i}, b_h^{\breve{l}_i}) + \overline{V}_{h+1}^{\breve{l}_i}(s_{h+1}^{\breve{l}_i})\right) \right] = 0, \quad (28)$$

which holds simply due to the martingale property. Finally, combining the results in (23), (24), (26), (28), we obtain that

$$
\begin{aligned}
\mathbb{E}\left[V_h^{\star,\bar{\nu}_h^k}(s)\right] &\leq \mathbb{E}\left[\frac{1}{\check{n}}\sum_{i=1}^{\check{n}}\max_{\mu}\mathbb{D}_{\mu\times\nu_h^{\check{l}_i}}\left(r_h + \mathbb{P}_h\overline{V}_{h+1}^{\check{l}_i}\right)(s)\right], \\
&\leq \mathbb{E}\left[\frac{1}{\check{n}}\sum_{i=1}^{\check{n}}\mathbb{D}_{\mu_h^{\check{l}_i}\times\nu_h^{\check{l}_i}}\left(r_h + \mathbb{P}_h\overline{V}_{h+1}^{\check{l}_i}\right)(s)\right] + 16H\sqrt{2A_{\max}^3/\check{n}^{1/4}} \\
&= \mathbb{E}\left[\frac{1}{\check{n}}\sum_{i=1}^{\check{n}}\left(r_h(s, a_h^{\check{l}_i}, b_h^{\check{l}_i}) + \overline{V}_{h+1}^{\check{l}_i}(s_{h+1}^{\check{l}_i})\right)\right] + 16H\sqrt{2A_{\max}^3/\check{n}^{1/4}} \\
&\leq \mathbb{E}\left[\frac{1}{\check{n}}\sum_{i=1}^{\check{n}}\left(r_h(s, a_h^{\check{l}_i}, b_h^{\check{l}_i}) + \overline{V}_{h+1}^{\check{l}_i}(s_{h+1}^{\check{l}_i})\right)\right] + b_{\check{n}} \\
&= \mathbb{E}[\overline{V}_h^k(s)],
\end{aligned}
$$

where the second to last step is by the definition of $b_{\check{n}} = 25H\sqrt{A_{\max}^3/\check{n}^{1/4}}$. In the last step we used the formulation of $\overline{V}_h^k(s)$ in (22).

**Case 2:** $\overline{V}_h(s)$ was not updated in (the end of) episode $k-1$. Since we have excluded the case that $\overline{V}_h$ has never been updated, we are guaranteed that there exists an episode $j$ such that $\overline{V}_h(s)$ has been updated in the end of episode $j-1$ most recently. In this case, $\mathbb{E}[\overline{V}_h^k(s)] = \mathbb{E}[\overline{V}_h^{k-1}(s)] = \cdots = \mathbb{E}[\overline{V}_h^j(s)] \geq \mathbb{E}[V_h^{\star,\bar{\nu}_h^j}(s)]$, where the last step is by the induction hypothesis. Finally, observe that by our definition, the value of $V_h^{\star,\bar{\nu}_h^i}(s)$ is a constant for all episode indices $i$ that belong to the same stage. Since we know that episode $j$ and episode $k$ lie in the same stage, we can conclude that $\mathbb{E}[V_h^{\star,\bar{\nu}_h^k}(s)] = \mathbb{E}[V_h^{\star,\bar{\nu}_h^j}(s)] \leq \mathbb{E}[\overline{V}_h^k(s)]$.

Combining the two cases completes our proof.

### G.2 PROOF OF THEOREM 3

In the following, we present the proof for the first inequality. The second bound can be proved via a similar argument.

Before proceeding further, we first recall the definitions of several notations from Section 5 and define a few new ones. For a state $s_h^k$, let $n_h^k$ be the total number of episodes that this state has been visited (at the $h$-th step) prior to the current stage, and let $l_{h,i}^k$ denote the index of the episode that this state was visited the $i$-th time among the total $n_h^k$ times. Similarly, recall that $\check{n}_h^k$ denotes the number of visits to the state $s_h^k$ in the stage right before the current stage, and $\check{l}_{h,i}^k$ denotes the $i$-th episode among the $\check{n}_h^k$ episodes. For simplicity, we use $l_i$ and $\check{l}_i$ to denote $l_{h,i}^k$ and $\check{l}_{h,i}^k$, and $\check{n}$ to denote $\check{n}_h^k$, whenever $h$ and $k$ are clear from the context.

From Lemma 1, we first obtain that

$$
\sum_{k=1}^K \mathbb{E}\left[V_1^{\star,\bar{\nu}_1^k}(s_1) - V_1^{\bar{\mu}_1^k,\bar{\nu}_1^k}(s_1)\right] \leq \sum_{k=1}^K \mathbb{E}\left[\overline{V}_1^k(s_1) - V_1^{\bar{\mu}_1^k,\bar{\nu}_1^k}(s_1)\right].
$$

We hence only need to upper bound the RHS. For ease of exposition, we define the following notation:

$$
\delta_h^k \stackrel{\text{def}}{=} \overline{V}_h^k(s_h^k) - V_h^{\bar{\mu}_h^k,\bar{\nu}_h^k}(s_h^k),
$$

The main idea of the proof is to upper bound $\sum_{k=1}^K \delta_h^k$ by the next step $\sum_{k=1}^K \delta_{h+1}^k$, and then obtain a recursive formula. From the update rule of $\overline{V}_h^k(s_h^k)$ in (2), we know that

$$
\overline{V}_h^k(s_h^k) \leq \mathbb{I}[n_h^k = 0]H + \frac{1}{\check{n}}\sum_{i=1}^{\check{n}}\left(r_h(s, a_h^{\check{l}_i}, b_h^{\check{l}_i}) + \overline{V}_{h+1}^{\check{l}_i}(s_{h+1}^{\check{l}_i})\right) + b_{\check{n}},
$$

where the $\mathbb{I}[n_h^k = 0]$ term counts for the event that the optimistic value function has never been updated for the given state.

Let $\mathcal{F}_i$ be the $\sigma$-algebra generated by all the random variables before the $\check{l}_i$-th episode. It can be seen that $\{r_h(s, a_h^{\check{l}_i}, b_h^{\check{l}_i}) + V_{h+1}^{\bar{\mu}_{h+1}^{\check{l}_i}, \bar{\nu}_{h+1}^{\check{l}_i}}(s_{h+1}^{\check{l}_i})\}_{i=1}^{\check{n}}$ is a martingale with respect to the filtration $\{\mathcal{F}_i\}_{i=1}^{\check{n}}$. For any probability $p \in (0, 1]$, let $\iota = \log(2SKH/p)$. From the Azuma-Hoeffding inequality, it holds with probability at least $1 - p/(2SHK)$ that

$$
\begin{aligned}
V_h^{\bar{\mu}_h^k, \bar{\nu}_h^k}(s) =& \frac{1}{\check{n}} \sum_{i=1}^{\check{n}} \mathbb{D}_{\mu_h^{\check{l}_i} \times \nu_h^{\check{l}_i}} \left( r_h + \mathbb{P}_h V_{h+1}^{\bar{\mu}_{h+1}^{\check{l}_i}, \bar{\nu}_{h+1}^{\check{l}_i}} \right)(s) \\
\geq& \frac{1}{\check{n}} \sum_{i=1}^{\check{n}} \left( r_h(s, a_h^{\check{l}_i}, b_h^{\check{l}_i}) + V_{h+1}^{\bar{\mu}_{h+1}^{\check{l}_i}, \bar{\nu}_{h+1}^{\check{l}_i}}(s_{h+1}^{\check{l}_i}) \right) - 2\sqrt{H^2 \iota/\check{n}},
\end{aligned}
\tag{29}
$$

where $\iota$ suppresses logarithmic terms. Substituting the two inequalities above into the definition of $\delta_h^k$, we have

$$
\begin{aligned}
\delta_h^k \leq& \mathbb{I}[n_h^k = 0]H + \frac{1}{\check{n}} \sum_{i=1}^{\check{n}} \left( r_h(s, a_h^{\check{l}_i}, b_h^{\check{l}_i}) + \overline{V}_{h+1}^{\check{l}_i}(s_{h+1}^{\check{l}_i}) \right) + b_{\check{n}} \\
& - \frac{1}{\check{n}} \sum_{i=1}^{\check{n}} \mathbb{D}_{\mu_h^{\check{l}_i} \times \nu_h^{\check{l}_i}} \left( r_h + \mathbb{P}_h V_{h+1}^{\bar{\mu}_{h+1}^{\check{l}_i}, \bar{\nu}_{h+1}^{\check{l}_i}} \right)(s) \\
\leq& \mathbb{I}[n_h^k = 0]H + \frac{1}{\check{n}} \sum_{i=1}^{\check{n}} \left( \overline{V}_{h+1}^{\check{l}_i}(s_{h+1}^{\check{l}_i}) - V_{h+1}^{\bar{\mu}_{h+1}^{\check{l}_i}, \bar{\nu}_{h+1}^{\check{l}_i}}(s_{h+1}^{\check{l}_i}) \right) + b_{\check{n}} + 2\sqrt{H^2 \iota/\check{n}} \\
=& \mathbb{I}[n_h^k = 0]H + \frac{1}{\check{n}} \sum_{i=1}^{\check{n}} \delta_{h+1}^{\check{l}_i} + b_{\check{n}} + 2\sqrt{H^2 \iota/\check{n}},
\end{aligned}
\tag{30}
$$

where in the last step we used the definition of $\delta_{h+1}^{\check{l}_i}$. In fact, since we only need the results in Theorem 3 to hold in expectation rather than with high probability, we can simply replace (29) with

$$
\mathbb{E}\left[ V_h^{\bar{\mu}_h^k, \bar{\nu}_h^k}(s) \right] = \frac{1}{\check{n}} \sum_{i=1}^{\check{n}} \mathbb{E}\left[ r_h(s, a_h^{\check{l}_i}, b_h^{\check{l}_i}) + V_{h+1}^{\bar{\mu}_{h+1}^{\check{l}_i}, \bar{\nu}_{h+1}^{\check{l}_i}}(s_{h+1}^{\check{l}_i}) \right],
$$

by invoking the martingale property. Consequently, we can also derive an expectation form of (30) as follows:

$$
\mathbb{E}\left[ \delta_h^k \right] \leq \mathbb{E}\left[ \mathbb{I}[n_h^k = 0]H + \frac{1}{\check{n}} \sum_{i=1}^{\check{n}} \delta_{h+1}^{\check{l}_i} + b_{\check{n}} \right].
\tag{31}
$$

To find an upper bound of $\sum_{k=1}^K \delta_h^k$, we proceed to upper bound each term in the RHS of (31) separately. First, notice that $\sum_{k=1}^K \mathbb{I}\left[ n_h^k = 0 \right] \leq SH$, because each fixed state-step pair $(s, h)$ contributes at most 1 to $\sum_{k=1}^K \mathbb{I}\left[ n_h^k = 0 \right]$. Next, we turn to analyze the second term in the RHS of (31). Observe that

$$
\begin{aligned}
\sum_{k=1}^K \frac{1}{\check{n}_h^k} \sum_{i=1}^{\check{n}_h^k} \delta_{h+1}^{\check{l}_{h,i}^k} &= \sum_{k=1}^K \sum_{j=1}^K \frac{1}{\check{n}_h^k} \delta_{h+1}^j \sum_{i=1}^{\check{n}_h^k} \mathbb{1}\left[ \check{l}_{h,i}^k = j \right] \\
&= \sum_{j=1}^K \delta_{h+1}^j \sum_{k=1}^K \frac{1}{\check{n}_h^k} \sum_{i=1}^{\check{n}_h^k} \mathbb{1}\left[ \check{l}_{h,i}^k = j \right].
\end{aligned}
\tag{32}
$$

For a fixed episode $j$, notice that $\sum_{i=1}^{\check{n}_h^k} \mathbb{1}[\check{l}_{h,i}^k = j] \leq 1$, and that $\sum_{i=1}^{\check{n}_h^k} \mathbb{1}[\check{l}_{h,i}^k = j] = 1$ happens if and only if $s_h^k = s_h^j$ and $(j, h)$ lies in the previous stage of $(k, h)$ with respect to the state-step pair $(s_h^k, h)$. Define $\mathcal{K}_j \overset{\text{def}}{=} \{k \in [K] : \sum_{i=1}^{\check{n}_h^k} \mathbb{1}[\check{l}_{h,i}^k = j] = 1\}$. We then know that all episode indices $k \in \mathcal{K}_j$ belong to the same stage, and hence these episodes have the same value of $\check{n}_h^k$. That is, there

exists an integer $N_j > 0$, such that $\check{n}_h^k = N_j, \forall k \in \mathcal{K}_j$. Further, since the stages are partitioned in a way such that each stage is at most $(1 + \frac{1}{H})$ times longer than the previous stage, we know that $|\mathcal{K}_j| \leq (1 + \frac{1}{H})N_j$. Therefore, for every $j$, it holds that

$$\sum_{k=1}^{K} \frac{1}{\check{n}_h^k} \sum_{i=1}^{\check{n}_h^k} \mathbb{1}\left[\check{l}_{h,i}^k = j\right] \leq 1 + \frac{1}{H}. \tag{33}$$

Combining (32) and (33) leads to the following upper bound of the second term in (31):

$$\sum_{k=1}^{K} \frac{1}{\check{n}_h^k} \sum_{i=1}^{\check{n}_h^k} \delta_{h+1}^{\check{l}_{h,i}^k} \leq (1 + \frac{1}{H}) \sum_{k=1}^{K} \delta_{h+1}^k. \tag{34}$$

So far, we have obtained the following upper bound:

$$\mathbb{E}\left[\sum_{k=1}^{K} \delta_h^k\right] \leq SH^2 + (1 + \frac{1}{H})\mathbb{E}\left[\sum_{k=1}^{K} \delta_{h+1}^k\right] + \sum_{k=1}^{K} b_{\check{n}_h^k}.$$

Iterating the above inequality over $h = H, H-1, \ldots, 1$ leads to

$$\mathbb{E}\left[\sum_{k=1}^{K} \delta_1^k\right] \leq O\left(SH^3 + \sum_{h=1}^{H} \sum_{k=1}^{K} (1 + \frac{1}{H})^{h-1} b_{\check{n}_h^k}\right), \tag{35}$$

where we used the fact that $(1 + \frac{1}{H})^H \leq e$. In the following, we analyze the bonus term $b_{\check{n}_h^k}$ more carefully. Recall our definitions that $e_1 = H$, $e_{i+1} = \lfloor(1 + \frac{1}{H})e_i\rfloor$, $i \geq 1$, and $b_{\check{n}} = 25H\sqrt{A_{\max}^3/\check{n}^{1/4}}$. For any $h \in [H]$,

$$\sum_{k=1}^{K} (1 + \frac{1}{H})^{h-1} b_{\check{n}_h^k} \leq \sum_{k=1}^{K} (1 + \frac{1}{H})^{h-1} 25H \sqrt{\frac{A_{\max}^3}{\left(\check{N}_h^k(s_h^k)\right)^{1/4}}}$$

$$= 25H\sqrt{A_{\max}^3} \sum_{s \in \mathcal{S}} \sum_{j \geq 1} (1 + \frac{1}{H})^{h-1} e_j^{-\frac{1}{8}} \sum_{k=1}^{K} \mathbb{I}\left[s_h^k = s, \check{N}_h^k(s_h^k) = e_j\right]$$

$$= 25H\sqrt{A_{\max}^3} \sum_{s \in \mathcal{S}} \sum_{j \geq 1} (1 + \frac{1}{H})^{h-1} w(s,j) e_j^{-\frac{1}{8}},$$

where we define $w(s,j) \stackrel{\text{def}}{=} \sum_{k=1}^{K} \mathbb{I}\left[s_h^k = s, \check{N}_h^k(s_h^k) = e_j\right]$ for any $s \in \mathcal{S}$. If we further let $w(s) \stackrel{\text{def}}{=} \sum_{j \geq 1} w(s,j)$, we can see that $\sum_{s \in \mathcal{S}} w(s) = K$. For each fixed state $s$, we now seek an upper bound of its corresponding $j$ value, denoted as $J$ in what follows. Since each stage is $(1 + \frac{1}{H})$ times longer than its previous stage, we know that $w(s,j) = \sum_{k=1}^{K} \mathbb{I}\left[s_h^k = s, \check{N}_h^k(s_h^k) = e_j\right] = \lfloor(1 + \frac{1}{H})e_j\rfloor$ for any $1 \leq j \leq J$. Since $\sum_{j=1}^{J} w(s,j) = w(s)$, we obtain that $e_J \leq (1 + \frac{1}{H})^{J-1} \leq \frac{10}{1+\frac{1}{H}} \frac{w(s)}{H}$ by taking the sum of a geometric sequence. Therefore, by plugging in $w(s,j) = \lfloor(1 + \frac{1}{H})e_j\rfloor$,

$$\sum_{j \geq 1} (1 + \frac{1}{H})^{h-1} w(s,j) e_j^{-\frac{1}{8}} \leq O\left(\sum_{j=1}^{J} e_j^{\frac{7}{8}}\right) \leq O\left(w(s)^{\frac{7}{8}} H^{\frac{1}{8}}\right),$$

where in the second step we again used the formula of the sum of a geometric sequence. Finally, using the fact that $\sum_{s \in \mathcal{S}} w(s) = K$ and applying the Cauchy-Schwartz inequality, we have

$$\sum_{h=1}^{H} \sum_{k=1}^{K} (1 + \frac{1}{H})^{h-1} b_{\check{n}_h^k} = O\left(H^2 \sqrt{A_{\max}^3} \sum_{s \in \mathcal{S}} \sum_{j \geq 1} (1 + \frac{1}{H})^{h-1} w(s,j) e_j^{-\frac{1}{8}}\right)$$

$$\leq O\left(K^{\frac{7}{8}} H^{\frac{17}{8}} S^{\frac{1}{8}} A_{\max}^{\frac{3}{2}}\right). \tag{36}$$

Summarizing the results above leads to

$$\mathbb{E}\left[\sum_{k=1}^{K}\delta_1^k\right] \leq O\left(SH^3 + K^{\frac{7}{8}}H^{\frac{17}{8}}S^{\frac{1}{8}}A_{\max}^{\frac{3}{2}}\right).$$

When $K$ is large enough such that $K \geq SH^{3/7}$, the second term becomes dominant, and we obtain the desired result:

$$\sum_{k=1}^{K}\mathbb{E}\left[V_1^{\star,\bar{\nu}_1^k}(s_1) - V_1^{\bar{\mu}_1^k,\bar{\nu}_1^k}(s_1)\right] \leq \sum_{k=1}^{K}\mathbb{E}[\delta_1^k] \leq O\left(K^{\frac{7}{8}}H^{\frac{17}{8}}S^{\frac{1}{8}}A_{\max}^{\frac{3}{2}}\right).$$

This completes the proof of the theorem.

## G.3 PROOF OF THEOREM 4

The result essentially follows from the "regret to PAC guarantee" conversion given in Jin et al. (2018). Specifically, we know from Theorem 3 that

$$\sum_{k=1}^{K}\mathbb{E}\left[V_1^{\star,\bar{\nu}_1^k}(s_1) - V_1^{\bar{\mu}_1^k,\bar{\nu}_1^k}(s_1)\right] \leq C_1 K^{\frac{7}{8}}H^{\frac{17}{8}}S^{\frac{1}{8}}A_{\max}^{\frac{3}{2}} \tag{37}$$

for some $C_1$ that is independent of $K$. It is important to notice that

$$\mathbb{E}\left[V_1^{\star,\bar{\nu}_1^k}(s_1) - V_1^{\bar{\mu}_1^k,\bar{\nu}_1^k}(s_1)\right] \geq 0, \tag{38}$$

for each individual $k \in [K]$, because $V_1^{\star,\bar{\nu}_1^k}(s_1)$ is defined as the dynamic best response instead of the best fixed response in hindsight. Since the two agents uniformly draw the index $\kappa$ from $\{1, 2, \ldots, K\}$, we can conclude from (37) and (38) that for any realization of $\kappa$,

$$\mathbb{E}\left[V_1^{\star,\bar{\nu}_1^\kappa}(s_1) - V_1^{\bar{\mu}_1^\kappa,\bar{\nu}_1^\kappa}(s_1)\right] \leq 5C_1 H^{\frac{17}{8}}S^{\frac{1}{8}}A_{\max}^{\frac{3}{2}}/K^{\frac{1}{8}},$$

with probability at least $\frac{4}{5}$. To ensure $5C_1 H^{\frac{17}{8}}S^{\frac{1}{8}}A_{\max}^{\frac{3}{2}}/K^{\frac{1}{8}} \leq \varepsilon$, it suffices to require that $K \geq 5^8 C_1^8 H^{17} S A_{\max}^{12}/\varepsilon^8$. Similarly, if $K \geq 5^8 C_2^8 H^{17} S A_{\max}^{12}/\varepsilon^8$ for some constant $C_2$, then

$$\mathbb{E}\left[V_1^{\bar{\mu}_1^\kappa,\star}(s_1) - V_1^{\bar{\mu}_1^\kappa,\bar{\nu}_1^\kappa}(s_1)\right] \leq 5C_2 H^{\frac{17}{8}}S^{\frac{1}{8}}A_{\max}^{\frac{3}{2}}/K^{\frac{1}{8}}.$$

Therefore, if the agents run Algorithm 2 for at least $K = \Omega(H^{17}SA_{\max}^{12}/\varepsilon^8)$ episodes, the randomly drawn policy pair $(\bar{\mu}_1^\kappa, \bar{\nu}_1^\kappa)$ will be an $\varepsilon$-approximate NE in expectation with probability at least $\frac{3}{5}$ by taking a union bound. To improve the success probability to an arbitrary value $1-p$ for $p \in (0, \frac{2}{5})$, one can adopt the standard "boosting" technique in the literature of randomized algorithms (Mitzenmacher & Upfal, 2017) , by repeating the process for $O(\log \frac{1}{p})$ iterations, and then choosing the iteration with the smallest resulting regret. Finally, we remark that we have made no effort to optimize the sample complexity dependence on $H, S$, and $A_{\max}$, which can certainly be improved.

## G.4 V-LEARNING SGD WITH VARIANCE REDUCTION

**Theorem 7.** *(Informal). For any $\varepsilon > 0$, replace the* SGD-IX *subroutine in Algorithm 2 with a variance-reduced SGD method (Algorithm 4), and set the bonus term to be $b_{\check{n}} = \frac{CHA_{\max}^{6/11}(\ln \check{n})^{1/2}}{\check{n}^{1/3}\varepsilon^{1/3}}$ for some absolute constant $C$ instead. Suppose that the two agents run such an algorithm for $K$ episodes with $K = \widetilde{\Omega}(1/\varepsilon^4) \cdot poly(H, S, A_{\max})$, and uniformly sample an episode index $\kappa$ from the set $\{1, \ldots, K\}$. Then, with probability at least $\frac{3}{5}$, the policy pair $(\bar{\mu}_1^\kappa, \bar{\nu}_1^\kappa)$ is an $\varepsilon$-approximate Nash equilibrium in expectation.*

*Proof sketch.* The proof follows a similar procedure as in the proofs of Lemma 1 and Theorems 3 and 4. Due to their similarities, in the following, we only sketch some of the differences in the proofs. By setting the new bonus term to be $b_{\check{n}} = \frac{CHA_{\max}^{6/11}(\ln \check{n})^{1/2}}{\check{n}^{1/3}\varepsilon^{1/3}}$, one can verify that the proof of Lemma 1

still holds. That is, the optimistic value function $\overline{V}_h^k(s)$ (with a modified bonus term) is still a valid upper bound of both $V_h^{\star,\bar{\nu}_h^k}(s)$ and $V_h^{\bar{\mu}_h^k,\star}(s)$ in expectation. Everything up to (35) in the proof of Theorem 3 also holds. Starting from (35), we further deduce that

$$\sum_{h=1}^{H}\sum_{k=1}^{K}(1+\frac{1}{H})^{h-1}b_{\tilde{n}_h^k} \leq \widetilde{O}\left(K^{\frac{2}{3}}/\varepsilon^{\frac{1}{3}}\right) \cdot \mathrm{poly}(H, S, A_{\max}),$$

by following a similar argument as when we derived (36). Plugging the above inequality back to the proof of Theorem 3, we obtain that

$$\frac{1}{K}\sum_{k=1}^{K}\mathbb{E}\left[V_1^{\star,\bar{\nu}_1^k}(s_1) - V_1^{\bar{\mu}_1^k,\bar{\nu}_1^k}(s_1)\right] \leq \widetilde{O}(K^{-\frac{1}{3}}\varepsilon^{-\frac{1}{3}}) \cdot \mathrm{poly}(H, S, A_{\max}), \text{ and}$$

$$\frac{1}{K}\sum_{k=1}^{K}\mathbb{E}\left[V_1^{\bar{\mu}_1^k,\star}(s_1) - V_1^{\bar{\mu}_1^k,\bar{\nu}_1^k}(s_1)\right] \leq \widetilde{O}(K^{-\frac{1}{3}}\varepsilon^{-\frac{1}{3}}) \cdot \mathrm{poly}(H, S, A_{\max}).$$

Finally, repeating the proof idea of Theorem 4 leads to the desired result. $\qquad\square$

## G.5    A SAMPLE COMPLEXITY LOWER BOUND

To obtain a sample complexity lower bound of the problem, we consider a simple instance where either $\mathcal{A}$ or $\mathcal{B}$ is a singleton, i.e., $A = 1$ or $B = 1$. Learning an approximate NE in such a Markov team reduces to finding a near-optimal policy in a single-agent RL problem. Applying the regret lower bound of single-agent RL yields the following result for RL in Markov teams.

**Corollary 1.** *(Corollary of Jaksch et al. (2010)). For any algorithm, there exists a two-player Markov team problem that takes the algorithm at least $\Omega(H^3 S A_{\max}/\varepsilon^2)$ episodes to learn an $\varepsilon$-approximate Nash equilibrium.*

We remark that such a lower bound might be very loose. Reducing to a single-agent RL problem evades the strategic learning behavior of the agents and the non-stationarity that such behavior causes to the environment, which in our opinion are the central difficulties of learning in Markov teams. To derive a tighter lower bound in our decentralized setting, one should also utilize the additional constraint that each agent only has access to its local information, a factor that Corollary 1 apparently does not take into account. It is hence unsurprising that when comparing Theorem 4 with Corollary 1, we can see an obvious gap in the parameter dependence.

## G.6    DECENTRALIZED MARL IN SMOOTH TEAMS

In the following, we address an important subclass of Markov teams, named *smooth teams*, using the `V-learning SGD` algorithm. More importantly, we show that our algorithm can achieve near team-optimality, i.e., find the best Nash equilibrium, in such problems.

Smooth games were first introduced in Roughgarden (2009) to study the Price of Anarchy (POA) in normal-form games. A large class of games are covered as examples of smooth games, including congestion games and many forms of auctions (Roughgarden, 2009; Syrgkanis & Tardos, 2013). The notion of smoothness was later extended to learning in normal-form games (Syrgkanis et al., 2015; Foster et al., 2016) and cooperative Markov games (Radanovic et al., 2019; Mao et al., 2020). This concept essentially ensures that the game has a bounded POA, and hence *decentralized* no-regret learning dynamics can possibly converge to near-optimality.

Define $V^\star$ to be the team-optimal value function, i.e., $V_h^\star(s) = \max_{\pi\in\Pi} V_h^\pi(s)$ for any $h \in [H], s \in \mathcal{S}$, where $\Pi$ is the set of joint policies of the agents. We consider the following definition of a smooth Markov team:

**Definition 5.** *(Adapted from Radanovic et al. (2019)). For $\lambda \geq 0$ and $0 < \rho < 1$, an $N$-player Markov team is $(\lambda, \rho)$-smooth if there exists a policy profile $\pi^\star$ such that for any policy profile $\pi = (\pi^i, \pi^{-i})$:*

$$V_h^{\pi^{i\star},\pi^{-i}}(s) \geq \lambda \cdot V_h^\star(s) - \rho \cdot V_h^\pi(s), \forall i \in \mathcal{N}, s \in \mathcal{S}, h \in [H].$$

The $(\lambda, \rho)$-smoothness ensures that agent $i$ continues doing well by playing its optimal policy even when the other agents are using slightly sub-optimal policies. It immediately follows that the resultant policies of Algorithm 2 converge to a $\lambda/(1 + \rho)$ factor of the team-optimal NE at a rate of $O(K^{-1/8})$.

**Theorem 5.** Let $K = \Omega(1/\varepsilon^8) \cdot \text{poly}(H, S, A_{\max})$. In a $(\lambda, \rho)$-smooth team, the value of running the auxiliary policies $\{(\bar{\mu}_h^k, \bar{\nu}_h^k)\}_{h=1,k=1}^{H,K}$ satisfies

$$\frac{1}{K} \sum_{k=1}^{K} \mathbb{E}\left[ V_1^{\bar{\mu}_1^k, \bar{\nu}_1^k}(s_1) \right] \geq \frac{\lambda}{1 + \rho} V_1^\star(s_1) - \frac{1}{1 + \rho} O(K^{-\frac{1}{8}}).$$

*Proof.* From Theorem 3, it holds that

$$\frac{1}{K} \sum_{k=1}^{K} \mathbb{E}\left[ V_1^{\bar{\mu}_1^k, \bar{\nu}_1^k}(s_1) \right] \geq \frac{1}{K} \sum_{k=1}^{K} \mathbb{E}\left[ V_1^{\star, \bar{\nu}_1^k}(s_1) \right] - O(K^{-\frac{1}{8}})$$

$$\geq \frac{1}{K} \sum_{k=1}^{K} \left( \lambda \cdot V_1^\star(s_1) - \rho \cdot \mathbb{E}\left[ V_1^{\bar{\mu}_1^k, \bar{\nu}_1^k}(s_1) \right] \right) - O(K^{-\frac{1}{8}}),$$

where the second step is simply by the definition of smoothness. Rearranging the terms leads to the desired result. $\square$

Equivalently, such a result also holds when the agents uniformly sample an episode index $\kappa$ from $\{1, \ldots, K\}$ and then run $(\bar{\mu}_h^\kappa, \bar{\nu}_h^\kappa)$. In Theorem 5, we illustrated our results in the case of two agents, although one can easily verify that a similar result holds more generally for $N$-player smooth teams. Our approach significantly generalizes Radanovic et al. (2019) and Mao et al. (2020) in that we design natural update rules for both agents, who play symmetric roles in the self-play setting; the other two works only assign the algorithm to *one* agent, and have to *assume* that the policy of the other agent changes slowly.

## H  SIMULATIONS

In this section, we demonstrate the empirical performances of our algorithms. We start by evaluating SGD-IX (Algorithm 1) on a classic matrix team problem. Then, we evaluate V-learning SGD (Algorithm 2) on a Markov team problem, and compare its performances with various benchmarks.

### H.1  MATRIX TEAMS

First, to evaluate Algorithm 1, we use a classic matrix team example from the literature (Claus & Boutilier, 1998; Lauer & Riedmiller, 2000). Its reward table is reproduced in Table 1, where agent 1 is the row player, and agent 2 is the column player, both being maximizers. The action spaces of the agents are $\mathcal{A} = \{a_0, a_1, a_2\}$ and $\mathcal{B} = \{b_0, b_1, b_2\}$. There are three deterministic Nash equilibria in this team, among which two of them, $(a_0, b_0)$ and $(a_2, b_2)$, are team-optimal. It would be preferred that the agents not only learn a NE, but also settle on the same NE out of the two team-optimal ones.

|       | $b_0$ | $b_1$ | $b_2$ |
|-------|-------|-------|-------|
| $a_0$ | 10    | 0     | -10   |
| $a_1$ | 0     | 2     | 0     |
| $a_2$ | -10   | 0     | 10    |

| $s_0$ | $b_0$ | $b_1$ |
|-------|-------|-------|
| $a_0$ | -2    | 5     |
| $a_1$ | 2     | -2    |

| $s_1$ | $b_0$ | $b_1$ |
|-------|-------|-------|
| $a_0$ | 0     | 0     |
| $a_1$ | 0     | 0     |

Table 1: Reward table for the matrix team.    Table 2: Reward tables for the Markov team.

We run our SGD-IX algorithm on this task for $T = 5000$ rounds, and we set the step size $\alpha_t = 0.002$ and the implicit exploration parameter $\gamma = 0.001$. We evaluate our algorithm in terms of both the rewards it obtained and its $L^2$ equilibrium gap. Specifically, we define the $L^2$ equilibrium gap as the $L^2$ distance to a equilibrium point. For a pair of strategies $(\mu, \nu) \in \Delta(\mathcal{A}) \times \Delta(\mathcal{B})$, its $L^2$ equilibrium gap is defined as:

$$\text{Gap}(\mu, \nu) \overset{\text{def}}{=} \left\| \mu - \mu^\dagger(\nu) \right\|_2^2 + \left\| \nu - \nu^\dagger(\mu) \right\|_2^2, \tag{39}$$

where $\nu^\dagger(\mu)$ (resp. $\mu^\dagger(\nu)$) is the best response with respect to $\mu$ (resp. $\nu$), and $\|\cdot\|_2$ is the $L^2$ norm. The simulation results are presented in Figure 2. All results are averaged over 20 runs. Notice that we evaluate two sets of strategy trajectories: The "Current" strategy $(\mu_t, \nu_t)$ is the strategy pair used by Algorithm 1 at round $t$, while the "Average" strategy $(\bar{\mu}_t, \bar{\nu}_t)$ is the uniformly sampled strategy pair we obtain after running Algorithm 1 for $t$ rounds. Drawing an analogy to the Markov team setting, "Current" denotes the actual policies $\{(\mu_h^k, \nu_h^k)\}_{h=1,k=1}^{H,K}$ used at the step $(k, h)$ of Algorithm 2, while "Average" represents the auxiliary policies $\{(\bar{\mu}_h^k, \bar{\nu}_h^k)\}_{h=1,k=1}^{H,K}$ that we have constructed. In matrix teams, the auxiliary policy at round $t$ simply reduces to uniformly drawing a random time index $\tau$ from $\{1, \ldots, t\}$ and running the strategy pair $(\mu_\tau, \nu_\tau)$.

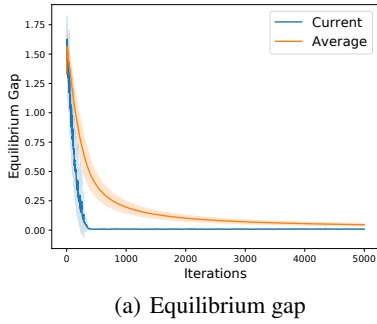

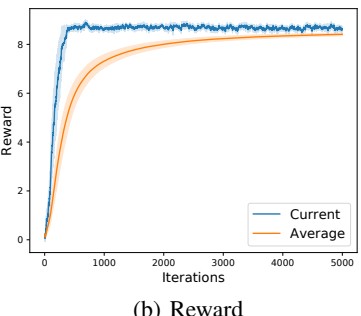

|(a) Equilibrium gap | (b) Reward |

Figure 2: (a) $L^2$ equilibrium gaps and (b) rewards of Algorithm 1 on the matrix team task given in Table 1. "Current" denotes the actual strategy trajectory, while "Average" represents the uniformly sampled strategy pair. Shaded areas denote the standard deviations of the equilibrium gap or reward.

Note that in Theorems 1 and 4 we only have theoretical guarantees for the auxiliary policies ("Average") but not the actual policy trajectories ("Current"). Encouragingly, our simulation results show that the actual policy trajectories also enjoy convergence behavior to NE empirically, and in most cases the "Current" policies converge even faster. Specifically, from Figure 2(a), we can see that the equilibrium gap of both "Current" and "Average" converge to zero, indicating that they indeed find an equilibrium as the number of iterations increase. The convergence of "Average" slightly lags behind "Current" because in Algorithm 1, "Average" simply takes the mean value over the actual trajectories, which requires some time to reflect the convergence behavior. A more promising result is that from Figure 2(b), we can see that the rewards collected by "Current" and "Average" converge to values close to 9. This suggests that Algorithm 1 not only finds a NE in this task, but actually converges to a team-optimal equilibrium most of the time. It does not exactly reach the team-optimal value of 10 because it still converges to non-team-optimal NE at a rather low frequency.

## H.2 MARKOV TEAMS

We further evaluate Algorithm 2 on a Markov team task. Inspired by Yongacoglu et al. (2019), we construct a Markov team problem with two states $\mathcal{S} = \{s_0, s_1\}$, where $s_0$ is the "good state" and $s_1$ is the "bad state". Each agent has two candidate actions $\mathcal{A} = \{a_0, a_1\}$ or $\mathcal{B} = \{b_0, b_1\}$. The reward function at each state is presented in Table 2. Specifically, at state $s_1$, both agents get a reward of 0 no matter what actions they select, while at state $s_0$, they will obtain a strictly positive reward if they either take the joint action $(a_0, b_1)$ or the one $(a_1, b_0)$. The state transition function is defined as follows:

$$P_h(s_0 \mid s_0 \text{ or } s_1, a_0, b_1) = 1 - \varepsilon, \ P_h(s_1 \mid s_0 \text{ or } s_1, \text{ not } (a_0, b_1)) = 1 - \varepsilon, \ \forall h \in [H],$$

and all the other transitions happen with probability $\varepsilon$. Intuitively, no matter which state the agents are in, they will transition to the good state $s_0$ with a high probability $1 - \varepsilon$ at the next step as long as they select the action pair $(a_0, b_1)$. All the other joint actions will lead to the bad state $s_1$ with a high probability $1 - \varepsilon$. The task hence rewards the agents who learn to consistently play the action pair $(a_0, b_1)$.

We run our `V-learning SGD` algorithm on this example for $K = 5000$ episodes, each episode containing $H = 10$ steps. We set the transition probability $\varepsilon = 0.1$, the step size $\alpha_t = 0.0004$, and the implicit exploration parameter $\gamma = 0.002$. We again evaluate our algorithm in terms of the reward and the equilibrium gap. The simulation results are presented in Figure 3, where all results are averaged over 20 runs. We would like to remind the reader that "Current" denotes the performances of the actual policy trajectory $\{(\mu_h^k, \nu_h^k)\}_{h=1,k=1}^{H,K}$, while "Average" represents the auxiliary policies $\{(\bar{\mu}_h^k, \bar{\nu}_h^k)\}_{h=1,k=1}^{H,K}$.

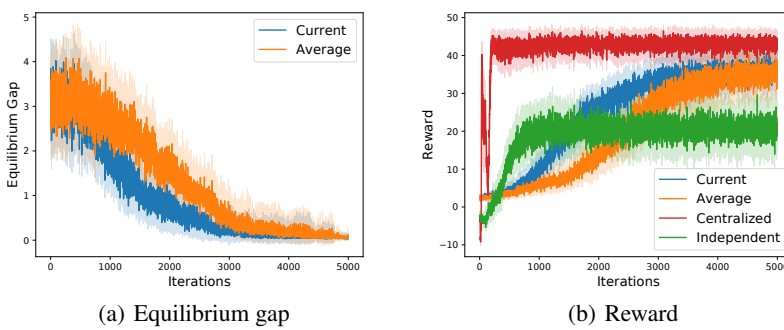

| (a) Equilibrium gap | (b) Reward |
|---|---|

Figure 3: (a) $L^2$ equilibrium gaps and (b) rewards of Algorithm 2 on the Markov team task given in Table 2. "Current" denotes the actual policy trajectory $\{(\mu_h^k, \nu_h^k)\}_{h=1,k=1}^{H,K}$, while "Average" represents the auxiliary policies $\{(\bar{\mu}_h^k, \bar{\nu}_h^k)\}_{h=1,k=1}^{H,K}$. Shaded areas denote the standard deviations of the equilibrium gap or reward.

We compare our algorithm with two meaningful benchmarks. The first benchmark is a "Centralized" oracle. This oracle acts as a centralized coordinator that can control the actions of both agents. Such an oracle essentially converts the Markov team task into a single-agent RL problem. The (randomized) action space of the centralized agent is $\Delta(\mathcal{A} \times \mathcal{B})$, which is larger than the $\Delta(\mathcal{A}) \times \Delta(\mathcal{B})$ space that we allow in our decentralized approach. Therefore, "Centralized" clearly upper bounds the performance that any decentralized learning algorithm can possibly achieve in this task. In our simulations, we implement "Centralized" by using a Hoeffding-based variant of a state-of-the-art single-agent RL algorithm UCB-ADVANTAGE (Zhang et al., 2020a). This algorithm has achieved the tightest sample complexity bound for single-agent RL in theory, and has also demonstrated remarkable empirical performances in practice (Mao et al., 2020). Such an algorithm could provide a strong performance upper bound in our task. The second benchmark we consider is the naïve "Independent" Q-learning. Specifically, we let each agent run a single-agent Q-learning algorithm independently, without being aware of the existence of the other agent or the structure of the Markov team. Each agent maintains an local optimistic Q-function, and takes greedy actions with respect to such optimistic estimates, without taking into account the other agents' actions. Since the agents update their policies simultaneously, the stationarity assumption of the environment in single-agent RL quickly collapses, and the theoretical guarantees for single-agent Q-learning no longer hold. This is also reminiscent of the "independent learner" approach proposed in an early work (Claus & Boutilier, 1998) for learning in Markov teams. We believe such a benchmark could provide meaningful intuitions about the consequences of not taking care of the team structure in decentralized methods. In our simulations, we implement such a benchmark by letting each agent running a variant of the single-agent UCB-ADVANTAGE (Zhang et al., 2020a) algorithm independently, where the (randomized) action spaces of the agents are $\Delta(\mathcal{A})$ and $\Delta(\mathcal{B})$.

We can observe from Figure 3(a) that the $L^2$ equilibrium gaps of both "Current" and "Average" converge to zero, indicating that both of them find a Nash equilibrium. This is encouraging because our theoretical results in Theorem 4 only guarantee the performance of the auxiliary policies, while our simulation results suggest that the actual policy trajectories also converge to a NE. The actual policy sequence also converges faster than the auxiliary policies, for reasons similar to the matrix team simulations. From Figure 3(b), we can again see that our algorithm even finds to the team-optimal NE most of the time, as the rewards it obtained nearly match the "Centralized" oracle. On the other hand,

the "Independent" benchmark converges, albeit faster, to a clearly suboptimal value. This reiterates that the naïve idea of independent learning does not work well for multi-agent RL in general, and a careful treatment of the game/team structure (like our SGD subroutine) is necessary. Finally, the implemented `V-learning SGD` algorithm takes much fewer samples to find an approximate NE than our theoretical results suggested. This indicates that the theoretical bounds might be overly conservative, and our algorithm could converge much faster in practice.

