# OpenReview forum: "Decentralized Cooperative Multi-Agent Reinforcement Learning with Exploration"
_ICLR.cc/2022/Conference — ICLR 2022 Submitted_

### Official Review · Reviewer_8WCV · 2021-10-20

**Correctness:** 2
**Technical Novelty And Significance:** 3
**Empirical Novelty And Significance:** 1
**Recommendation:** 3
**Confidence:** 4

**Main Review:**

There are the following problems of this paper.
- Typically, a Nash equilibrium is required to be a product policy (the players cannot have shared randomness). The policy pairs $(\mu_h^\kappa, \nu_h^\kappa)$ generated by Algorithm 3 is not a product policy, so this is not a Nash-equilibrium. In line 6, the authors propose to use a shared random variable $i$ among all the players (this is essential for the value iteration equation above Theorem 3 to hold; if the value iteration equation does not hold, the whole proof is wrong), and such shared randomness makes the policy correlated. Such a correlated policy is a CCE, but not Nash-equilibrium. The authors can argue that, by Markov inequality, the first step randomness $\kappa$ can be de-randomized, so theorem 1 and 2 may be valid (but these are just single step Markov games). But for Markov games, the later steps' randomness cannot be decoupled by the Markov inequality.

- Even if the issue can be fixed, the authors just considered two players Markov cooperative games, instead of multi-players Markov cooperative games. It is not clear whether it can be generalized to multi-players games with similar bounds.


Minor problems:
1. In the abstract the author wrote that their sample complexity is O(1/eps^4) for Markov team. However, it is O(1/eps^4) for Matrix team and O(1/eps^8) for Markov team.
2. The paper by Leonardos et.al. 2021 considered "the aspect of exploring the unknown environment" and had a sample complexity guarantee (in the related work section).




**Summary Of The Paper:**

This paper studies sample complexity bound for finding approximate Nash equilibrium in two-player Markov cooperative games using decentralized algorithms. The authors designed a stage-based V-learning algorithm, which achieves eps approximate Nash equilibrium in $1/eps^8$ episodes. Under the smoothness conditions, such equilibrium is a team-optimal Nash.



**Summary Of The Review:**

The main theorem is wrong.

---

> ### Author Response · Authors · 2021-11-23
> **Responses to Reviewer 8WCV**
>
> We thank the reviewer for the feedback. There seems to be some misunderstandings of the results. Our detailed responses are as follows.
> 1. We appreciate the reviewer’s concerns about the correctness of some of our statements, but we would like to refer the reviewer to the following facts that might help the reviewer better understand our results. It is known that V-learning with a no(-external)-regret learning algorithm leads to a coarse correlated equilibrium (CCE), and V-learning with no-swap-regret learning leads to a correlated equilibrium (CE) in general-sum Markov games (Jin et al., 2021). We would like to point out that our algorithm is evaluated based on the “dynamic regret”, which is a stronger concept than both (external) regret and swap regret. Such strong guarantees allow us to sequentially “de-randomize” the coupled policies. Specifically, given a realization of $\kappa$ and the realization of the sampling process thereafter, we indeed obtain a product policy (i.e., distributions over each agent’s individual action spaces, instead of over the joint action space). As an example, in the case of matrix games, it is known that the time-averaged policy (empirical frequency of plays) constitutes a CCE, but what our algorithm ensures is that a “certain step” of the policy pair (instead of the time-averaged policy) is a NE. Given the realization of such randomness, the policy pair is conditionally independent, and is no longer correlated through the common random seed. In other words, the correlated distribution $(\bar{\mu}_h^k,\bar{\nu}_h^k)$ itself is indeed a CCE, but the realization of the iterative sampling process is what we really obtain as a Nash equilibrium.
> 2. We state our results in the case of two-player teams only for ease of presentation. Our results directly generalize to multi-player teams with only one change that replaces $A_{\max}$ in the bound with the sum of the sizes of all players’ action spaces. This is because our algorithm does not have an explicit dependence on the number of agents. The reviewer can also easily verify that all our supporting lemmas (equivalence between stationary points and NE, convergence of decentralized SGD) still hold regardless of the number of players.
> 3. We would like to clarify that our $(1/\epsilon^4)$ sample complexity holds both for matrix teams and Markov teams. The $(1/\epsilon^4)$ bound in Markov teams is achieved in a similar way by incorporating the variance reduction techniques for SGD. Such a result is both stated in the main text and sketched in Theorem 7 of Appendix G.
> 4. The paper by (Leonardos et.al., 2021) indeed had a sample complexity guarantee, though strategic exploration was not the main focus of the paper.
>
> Reference:
>
> Jin, Chi, Qinghua Liu, Yuanhao Wang, and Tiancheng Yu. "V-Learning--A Simple, Efficient, Decentralized Algorithm for Multiagent RL." arXiv preprint arXiv:2110.14555 (2021).
>
> Ziang Song, Song Mei, and Yu Bai. "When Can We Learn General-Sum Markov Games with a Large Number of Players Sample-Efficiently?." arXiv preprint arXiv:2110.04184 (2021).

---

> > ### Comment · Reviewer_8WCV · 2021-11-24
> > **I still don't think the policy can be turned into a Nash Equilibrium**
> >
> > Thanks for the detailed response of the authors. I still don't think the policy can be turned into a Nash Equilibrium.
> >
> > In Algorithm 3, line 6, there are shared random variables $i$ for every state $s_h'$. At the initial state, this random variable $i$ is the $\kappa$. If any of these shared random variables $i$ for a state $s_h'$ is not pre-determined before the game, the policy is a correlated policy.
> >
> > To make the policy to be a NE, all these shared random variables $i$ should be derandomized (sampled before the game and keep them to be fixed). However, if one just samples them before the game, the value function associated with the derandomized policy is not simply the average of the value function associated with the correlated policy (the problem is that expectation and maximization cannot be exchanged). As a consequence, I don't see how the authors can turn the guarantee on the regret of the correlated policy (Theorem 3) to a guarantee that the derandomized policy is NE.
> >
> > The authors wrote "Such strong guarantees allow us to sequentially “de-randomize” the coupled policies." I didn't find in the paper where the authors discuss how random variables $i$ can be derandomized. Could the authors explain how this can be done?
> >
> > I can understand how the initial random variable $\kappa$ can be derandomized: using the Markov inequality. But I don't think such a method can be used to derandomize other shared random variables.

---

> > > ### Author Response · Authors · 2021-11-30
> > > **Further Responses to Reviewer 8WCV**
> > >
> > > We thank the reviewer very much for the detailed follow-up discussions. We are now clearer about the reviewer’s potential confusion as the reviewer has provided more details about his/her question. The reviewer is very correct in that “the value function associated with the de-randomized policy is not the average of the value function associated with the correlated policy”, because one cannot change the order of expectation and maximization. However, we would like to point out that our proof proceeds exactly in the way that no such “change-of-order” is required. We believe that the reviewer might want to double-check the difference between the “no-regret”-type guarantees in our work and the existing works in the literature. Specifically, it would be helpful to contrast our definition of $V_h^{\star, \bar{\nu}_h^k}$ (Equation (3) of page 7) with its counterparts in existing works, e.g., Equation (ii) in the proof of Lemma 16 in Bai et al., 2020 (zero-sum games), and the second equation in the proof of Lemma C.4 in Song et al., 2021 (learning CCE in general-sum games). The emphasis is that our max operator is “inside” the summation or “expectation”, while for the other two works the max is at the outside. Our criterion is comparing with the “best response” of each “individual realization” of the opponent’s policy, which we called a “dynamic regret guarantee” in our first round of response. It is in a stronger sense than what has been shown in existing works, and hence avoids the need for exchanging expectation with maximization to obtain a Nash equilibrium. Such a stronger result is achieved due to the convergence of SGD to stationary points (a nice property of the common-reward/team problem), and in general is not guaranteed for generic no-regret learning settings. This is indeed a subtle difference that deserves more discussions in the future version of this paper, and we hope that clarifications on such a nuance can help address the reviewer’s confusion. We really appreciate the expert-level questions and discussions from the reviewer.

---

> > > > ### Comment · Reviewer_8WCV · 2021-11-30
> > > > **Why "dynamic regret" help to "de-randomize" the correlated policy?**
> > > >
> > > > Thanks for the detailed response of the authors. I followed the authors' suggestions and looked at the difference with Song et. al. and Bai et. al.. I agree with the authors that the regret considered in this paper is a stronger notion of regret, and I see that the max operator is inside the expectation.
> > > >
> > > > However, I still don't understand why this "dynamic regret" helps to "de-randomize" and obtain product policies. From the rebuttal and the latest revision, I also can't find any further clue to "sequentially de-randomize the coupled policies“ as the authors said in the rebuttal.
> > > >
> > > > From my understanding, $(\bar\mu_1^k,\bar\nu_1^k)$ is a correlated policy for each $k$ because  $\bar\mu_1^k,\bar\nu_1^k$ use a common random seed to sample $i$ in Algorithm 3. The common random seed is also necessary for the correctness of Theorem 3 and Theorem 4 (so I think the statement of Theorem 3 and Theorem 4 needs to be revised).
> > > >
> > > > The authors should make it super clear how to "de-randomize" and come up with a theorem to explain why this works. Perhaps the authors think this is trivial: there may be a one-line proof of such a thing, which I am not able to come up with myself. But I think it should be the author's responsibility to write out such a one-line proof. Otherwise, I cannot see how such a claim is true.

---

> > > > > ### Author Response · Authors · 2021-12-01
> > > > > **Additional Response to Reviewer 8WCV**
> > > > >
> > > > > Thank you very much for the response. The dynamic-regret type of guarantee helps de-randomization in the sense that any one of the pairs of policies drawn from the $\check{n}$ pairs would correspond to a Nash (since our max is inside the summation). This follows the reasoning as the reviewer mentioned for de-randomization for "step 1" (i.e., from 1 to K). The common random seed is only used to "pick" the "matching pair" of the policies, and one way to do it is to sample the random variables before the game, as the reviewer suggested. Note that for any such "picked" policy pair, they are product policies, i.e., there is no correlation between choosing actions for player 1 in action set $\mathcal{A}$, and for player 2 in action set $\mathcal{B}$. Thanks again for the suggestion, and we will make it more explicit in the next version.

---

> > > > > > ### Comment · Reviewer_8WCV · 2021-12-01
> > > > > > **Why this de-randomized policy is Nash? Can you write down a theorem statement and sketch a proof?**
> > > > > >
> > > > > > I understand the "derandomized" policy, and understand that this is a product policy. But why this policy is a Nash? Could the authors write down a theorem *giving a high probability statement on the value function of the "derandomized" policy and show that this statement guarantees that the "derandomized" policy is a Nash*, and then sketch a proof for such a theorem? Note that the value function of the "derandomized policy" in this theorem should be expectation-free with respect to the "shared randomness". The main theorems in the paper claim that they are "Nash in expectation". I don't quite understand the meaning of "Nash in expectation", and would hope a high probability statement.

---

### Official Review · Reviewer_iszg · 2021-11-01

**Correctness:** 3
**Technical Novelty And Significance:** 3
**Empirical Novelty And Significance:** 3
**Recommendation:** 6
**Confidence:** 2

**Main Review:**

Pros
+ The paper is theoretically strong. Theorem 4, shows a 60% convergence probability of their algorithm on expectation, yet with two more iterations (if I did the math right) it bumps it to 99%.
+ The sample complexity reduction compared to the previous work is impressive.

Room for Improvement
- The paper is difficult to follow. I recommend pushing more sections into the Appendix so you can explain the existing content well in the main body. For example, Algorithm 2 was mentioned without enough discussion. How certain constants where picked, etc.
- Given the strong promise of the paper, I was underwhelmed with the simulation results. Why did you pick a simple 2 player game? Why not going for something much larger? After all that is the main catch of your bound.
- For the simulation result, I think you could also calculate he optimal policy for the centralized case using backward dynamic programming in time since your H is fixed. How come you did not include that?

Details
- "Oblivious to the presence of agents" => Does this limit the applicability of the method? For example if multiple agents are trying to do something together, often their location would heavily impact each other.
- O(1/e^4) samples (paper) or episodes (abstract)? While given the fixed H and Poly part of the theorem it does not matter, it would be great to make ti consistent.
- Any thoughts of expanding this to infinite horizon problem?
- recommend using t instead of h for tilmestep.

**Summary Of The Paper:**

Authors tackles the cooperative multi-agent reinforcement learning problem with stochastic dynamics. They introduced V-learning Stochastic Gradient Descent (SGD) with theoretical proofs of convergence to the optimal policy with desired e probability. The resulting sample complexity is O(1/e^4) * Poly (A,S,H), where H is the episode horizon. Authors also empirically tested the algorithm in a toy 2 agent matrix game showing convergence to the optimal policy.

**Summary Of The Review:**

I think the theoretical contributions of the paper are great and would benefit the community. I was hoping to see much bigger domain in the simulation part. The writing is currently very dense.

---

> ### Author Response · Authors · 2021-11-23
> **Responses to Reviewer iszg**
>
> We thank the reviewer for the valuable comments. Following the suggestion of the reviewer, we have carefully revised the writing of the analytical parts to allow for more explanation and intuition. Our detailed responses are as follows.
> 1. Given the reviewer’s interest in our experimental performance, we would like to let the reviewer know that we have also added new numerical results during the rebuttal period. We conducted simulations on a classic DecPOMDP task named Boxpushing (Seuken and Zilberstein, 2007) with a larger state space (proper modifications are made to be consistent with our formulation), as well as a variant of the GoodState task with more agents. The new empirical results are attached as supplementary material.
> 2. In the simulation task, the optimal value calculated from backward dynamic programming is 44.34, which is essentially equal to what the “Centralized” baseline obtained. We have revised the manuscript accordingly to make this point clear.
> 3. “Oblivious to the presence of agents”: We would more like to consider this as an advantage instead of a limitation of our method. An “oblivious” method can still go through even in the case where information sharing is possible, while a method for the latter scenario might not longer work in the former decentralized setting because it requires more information than what the environment can provide. Our decentralized condition in this sense relaxes the informational requirement of the algorithm, instead of intensifying it. The reviewer is definitely correct that when such information sharing is indeed possible, algorithms that are designed to specifically rely on such information sharing could achieve higher values, because they search for the optimal policy in a larger policy space. Designing algorithms for such scenarios is another interesting research direction, but it induces a different problem formulation, and is not the focus of our work (i.e., developing decentralized algorithms in the setting when such information sharing is not available).
> 4. “O(1/e^4) samples (paper) or episodes (abstract)?”: We thank the reviewer for pointing out this inconsistency. We have updated the draft accordingly.
> 5. Extending to the infinite horizon is absolutely an important research question. We believe that it would be helpful to adopt the exploration techniques designed for single-agent RL in the infinite-horizon scenario (Wang et al., 2019). But we would also expect some difficulties in the multi-agent setting because the auxiliary/certified policies that we/others have constructed are “non-Markov” policies, whose counterparts are not clear in the infinite-horizon case as far as we are aware of. We have left addressing this infinite-horizon setting as one important work for the future.
>
> Reference:
>
> Sven Seuken, and Shlomo Zilberstein. "Improved memory-bounded dynamic programming for decentralized POMDPs." Proceedings of the Twenty-Third Conference on Uncertainty in Artificial Intelligence. 2007.
>
> Yuanhao Wang, Kefan Dong, Xiaoyu Chen, and Liwei Wang. "Q-learning with UCB Exploration is Sample Efficient for Infinite-Horizon MDP." In International Conference on Learning Representations. 2019.

---

> > ### Comment · Reviewer_iszg · 2021-11-27
> > **Empirical evaluation**
> >
> > Thank you for expanding the work to a new domain. Although the new domain still feels like a tiny domain. Can you elaborate a bit more on the limitations of scaling it to a much larger setting?

---

> > > ### Author Response · Authors · 2021-11-30
> > > **Further Responses to Reviewer iszg**
> > >
> > > We thank the reviewer for the follow-up discussion. The new Boxpushing task that we evaluated on is an environment with ~100 states, where each agent has 4 actions. The main reason that prevents us from evaluating on larger-scale tasks is that our theoretical guarantees only cover the tabular case (finite states and actions), while significantly larger environments generally require function approximations / neural networks in the solutions, in order to make the algorithm converge in reasonable amount of time. Nevertheless, we believe that our stage-based learning paradigm can be helpful when combined with deep RL in practice, even though our theory does not cover such cases. We believe another possible limitation is that larger-scale tasks generally have more suboptimal Nash equilibria. Since our method only guarantees finding a NE but not necessarily a team-optimal one, it might more easily get stuck at local optima in those tasks. We would also like to remark that given fixed state and action sets, the computation complexity that most people are concerned about is actually not a main issue that limits our scalability (in terms of being with respect to the number of agents, and the computation required for each step). For instance, our method does not require much more computation than independent Q-learning, and we have not experienced any computational bottleneck when simulating our method. We hope our response has summarized some of the limitations, and will add these remarks in the next version of the paper. We appreciate the reviewer’s helpful comment on this point.

---

### Official Review · Reviewer_t9pe · 2021-11-02

**Correctness:** 1
**Technical Novelty And Significance:** 4
**Empirical Novelty And Significance:** 4
**Recommendation:** 8
**Confidence:** 2

**Main Review:**

The paper is extraordinarily well written; I did not find any issues with spelling, grammar nor presentation.
The authors provide a variety of interesting theoretical results.

The O(1/epsilon^4) result from Theorem 2 is particularly exciting. An extension to continuous action spaces (with some reasonable structural assumptions) would be interesting.

The use of a stage-based approach to tackle multi-agent learning non-stationarity is by itself refreshing (even without the perks of establishing unique non-asymptotic convergence bounds), and I expect it will resonate within the deep MARL community.

In section 4, I would suggest using a letter other than \mathcal{L} for denoting the partial sums over stage lengths, as \mathcal{L} is frequently used to denote loss objectives in MARL literature.

To further improve the paper, I believe the authors should compare their empirical results against other methods that aim to tackle nonstationarity in multi-agent learning. These include CTDE algorithms, such as tabular variants of QMIX [Rashid et al] or VDN [Sunehag et al]. This also includes tabular variants of policy-gradient algorithms, such as IPPO [Schroeder de Witt et al].

I see two major opportunities for future work:
1. Extend V-learning SGD to Q-learning with function approximation, ideally with theoretical guarantees, but an empirical demonstration would also be extremely interesting
2. Can V-learning SGD additionally benefit from CTDE settings, i.e. can the availability of extra state information during training be exploited? What about parameter-sharing in function approximation settings?

**Summary Of The Paper:**

The authors establish a variety of novel theoretical results for Markov team problems. First of all, they present a novel non-asymptotic convergence bound for single-step Markov teams, using a no-regret independent learning variant of stochastic gradient descent (SGD-IX). Secondly, they extend SGD-IX with a stage-based design to arrive at a novel, decentralized MARL algorithm (Stage-based V-learning). For Stage-based V-learning, the authors additionally proceed to establish a novel non-asymptotic convergence bounds that do not depend on the size of the centralised joint policy, but instead just the largest individual action space. The authors also show how their algorithm sidesteps technical difficulties inherent to establishing convergence bounds for Optimistic Q-learning [Jin et al, 2018]. Using a simple matrix team task, as well as a Markov team task as an example, the authors demonstrate empirically that their algorithms can end up selecting favourable Nash equilibria.

**Summary Of The Review:**

I recommend the paper be accepted as is. The paper's writing and presentation are very clear and polished, and the results are both sound and significant. I expect this paper to stimulate the cooperative MARL community.

---

> ### Author Response · Authors · 2021-11-23
> **Responses to Reviewer t9pe**
>
> We thank the reviewer for the appreciation of our paper and the helpful comments. Our detailed responses are as follows.
>
> 1. We thank the reviewer for pointing out the interesting future directions for this work. We fully agree that incorporating V-learning SGD with function approximation would be an important future direction, both theoretically and practically. We hope that our results on the convergence of stage-based learning can also inspire exciting research ideas in the deep MARL community. As a theory-oriented work, we have not considered specific training paradigms such as CTDE or parameter sharing, but making connections to these popular empirical approaches would definitely be valuable and hopefully bring theoretical justifications for their empirical successes in DecPOMDPs.
> 2. We thank the reviewer for pointing out the potential notational ambiguity of $\mathcal{L}$. Proper modifications have been made in the paper.
> 3. Given the reviewer’s interest in our experimental performance, we would like to let the reviewer know that we have also added new numerical results during the rebuttal period. We conducted simulations on a classic DecPOMDP task named Boxpushing (Seuken and Zilberstein, 2007) with a larger state space (proper modifications are made to be consistent with our formulation), as well as a variant of the GoodState task with more agents. The new empirical results are attached as supplementary material.
> 4. Given the limited time we have for the rebuttal period, we were unfortunately unable to complete the numerical comparisons with existing empirical methods as the reviewer suggested, such as QMIX and VDN. Reducing these methods to the tabular case also requires some special care to ensure a fair comparison. We believe that such comparisons would be beneficial to the community, and we leave them to the future version of this work.
>
> Reference:
>
> Sven Seuken, and Shlomo Zilberstein. "Improved memory-bounded dynamic programming for decentralized POMDPs." Proceedings of the Twenty-Third Conference on Uncertainty in Artificial Intelligence. 2007.

---

### Official Review · Reviewer_hhmV · 2021-11-04

**Correctness:** 3
**Technical Novelty And Significance:** 3
**Empirical Novelty And Significance:** 3
**Recommendation:** 6
**Confidence:** 4

**Main Review:**

This is an interesting subclass of problem, I found the paper was well written and explained the approach clearly. The authors correctly point out the considerable scaling issues with communication in agent based cooperative tasks that attempt to exchange information between the agents, and since it is easy for those to be intractable, it is very useful to identify alternatives.

I take mild issue with the claim that there is "no communication" between agents in this kind of approach though - there is, it is the information distributed to all the agents at the beginning of the algorithm in the form of the reward function. The agents then don't rely on any further exchanges of information to achieve a local optimum that achieves a global approximate Nash equilibrium policy. This in no way discredits the approach, and there is an interesting sub-class of problems that can be solved this way (for example American Football´s play books somewhat fit this model), but I would like to see this pointed out in the discussion. There is a danger, especially in simulating this problem, that the reward function contains enough information to in some sense encode the solution, and It would improve the paper significantly if there was a discussion of the size of the information communicated by the reward function, versus the settling time of the algorithm. It would also be nice to see concrete real world examples where this approach is useful, although I appreciate that this is a theoretical paper, making that in some sense somebody else's job.

**Summary Of The Paper:**

This paper examines the problem of completely decentralised agent cooperation in a Markov game environment, where there is no communication between agents. They propose a stage based learning algorithm to explore an unknown environment using this approach.

**Summary Of The Review:**

I think this paper is a useful contribution, and although I don't think the issues I raise are completely minor, they shouldn't be too hard to address by the authors.

---

> ### Author Response · Authors · 2021-11-23
> **Responses to Reviewer hhmV**
>
> We thank the reviewer for the helpful feedback. We totally agree that the scalability issue due to communications can be critical in multi-agent RL, and we believe that our decentralized learning paradigm can offer a useful alternative solution.
>
> We would also like to reassure the reviewer that the “information distributed through the reward function” is only a minor issue in our approach. In (single/multi-agent) RL theory research, it is commonly assumed that the reward function is deterministic or is distributed to the agent(s) a priori (Jin et al., 2018, Bai et al., 2020). This is motivated by the observation that transitions are generally more difficult to learn than the reward function, and such an assumption does not change the leading term of the sample complexity bound. We assumed a similar condition in our paper mainly for ease of presentation. We would like to point out that our approach directly generalizes to the case where such an assumption is absent; that is, all our results still hold even if the reward function is not given to the agents at all. In fact, the agents in our approach do not make strategic use of the reward to achieve any implicit communication, and our solution focuses on the general scenario that in no way the information/solution can be encoded in the reward function. An agent can proceed by observing only the realization of a scalar-valued reward at each step (instead of the reward function per se), without even knowing the existence of the other agents. In our simulations, the rewards are also only used to perform gradient updates on each agent’s policy individually, without relying on any sophisticated information exchange among the agents.
>
> Given the reviewer’s interest in our experimental performance, we would like to let the reviewer know that we have also added new numerical results during the rebuttal period. We conducted simulations on a classic DecPOMDP task named Boxpushing (Seuken and Zilberstein, 2007) with a larger state space (proper modifications are made to be consistent with our formulation), as well as a variant of the GoodState task with more agents. The new empirical results are attached as supplementary material.
>
> Reference:
>
> Sven Seuken, and Shlomo Zilberstein. "Improved memory-bounded dynamic programming for decentralized POMDPs." Proceedings of the Twenty-Third Conference on Uncertainty in Artificial Intelligence. 2007.

---

### Decision · Program_Chairs · 2022-01-20

**Decision:**

Reject

**Comment:**

This paper presents a decentralized cooperative approach in multi-agents using Markov games theory. After reviewing the paper and reading the comments from the reviewers, here are my comments:

- The paper is well-written, quite difficult to follow, but very informative.
- The contribution is clearly stated and the results support it.
- Theoretical results are interesting for the RL community.
- The main concern is about learning the epsilon-approximate Nash equilibrium policy which is a fundamental part of the paper.